# RETHINKING GRAPH-BASED DOCUMENT CLASSIFICATION: LEARNING DATA-DRIVEN STRUCTURES BEYOND HEURISTIC APPROACHES

## ABSTRACT

In document classification, graph-based models effectively capture document structure and overcome sequence length limitations, enhancing contextual understanding. However, existing graph document representations often rely on heuristics, domain-specific rules, or expert knowledge. We propose a novel method to learn data-driven graph structures, eliminating the need for manual design and reducing domain dependence. Our approach constructs homogeneous weighted graphs with sentences as nodes, while edges are learned via a self-attention model that identifies dependencies between sentence pairs. A statistical filtering strategy retains only strongly correlated sentences, improving graph quality while reducing the graph size. Experiments[1] on three datasets show that learned graphs consistently outperform heuristic-based baselines and recent small language models, achieving higher accuracy and $F_1$ score. Furthermore, our study demonstrates the effectiveness of the statistical filtering in improving classification robustness, highlighting the potential of automatic graph generation over traditional heuristic approaches and opening new directions for broader applications in NLP.

## 1 INTRODUCTION

Traditional vector-based text representation methods often struggle to effectively capture the structural information inherent in text, particularly when dealing with long documents. In contrast, graph-based representations have emerged as a powerful alternative, enabling the modeling of dependencies between textual units and leveraging their structure to better capture and differentiate local contexts within a document. Such representations have demonstrated promising results in document classification tasks (Zhang et al., 2020; Wang et al., 2023; Gu et al., 2023; Li et al., 2025b), with various graph construction strategies proposed to date.

However, existing graph-based approaches heavily rely on domain-specific heuristics and expert knowledge, limiting their generalizability across tasks. As noted by Wang et al. (2024), graph structures in text classification are typically implicit and require manual design tailored to each application. Consequently, these methods are typically effective only within narrow, predefined scenarios (Bugueño & de Melo, 2023; Galke & Scherp, 2022). To reduce the reliance on manually defined heuristics, a more robust and adaptive approach is needed.

While attention mechanisms have been widely adopted to model long-range dependencies, their usage to induce latent relational structure remains underexplored in document classification. In this work, we view attention scores as potential evidence for semantic ties between textual units, serving as a data-driven proxy for graph topology. Through statistical filtering, we derive sparse, task-conditioned relational structures that adapt to a document's internal organization rather than reflecting externally imposed heuristics.

We introduce a self-attention-based framework that, to our knowledge, is the first to automatically learn graph structures for document representations without handcrafted heuristics (see Figure 1). Our method builds homogeneous graphs where nodes represent sentences and edges are determined

---

[1]https://github.com/available/upon/publication

Figure 1: Instead of using domain-specific heuristics for graph construction, we learn graph structures from text, eliminating task-specific design and enhancing generalization.

by an attention model that learns relationships between sentence pairs. A statistical filtering step–using mean- or max-bound thresholds derived from the learned weight distribution–retains only the most salient relationships.

We evaluate our approach on three document classification datasets of varying lengths, comparing our learned graphs to five widely used heuristic-based construction strategies–complete graph, sentence order (Castillo et al., 2015; Bugueño & de Melo, 2023), window-based co-occurrence (Hassan & Banea, 2006; Rousseau et al., 2015; Zhang et al., 2020; Li et al., 2025b), and semantic similarity under predefined thresholds (Li et al., 2025b; Mihalcea & Tarau, 2004; Bugueño et al., 2024)–as well as competitive transformer and embedding-based models (Section 4.2). The results reveal that attention-learned graphs consistently outperform heuristic graphs and recent non-graph baselines, with gains becoming more pronounced as the document length increases. Further analysis finds that max-bound filtering is most effective for long documents, while mean-bound filtering fares better on medium-length documents. These findings highlight the potential of data-driven graph learning over conventional heuristic approaches and open new directions for broader applications.

The key contributions of this paper are:

- A novel data-driven graph generation model: We introduce a self-attention-based approach that learns sentence-level graph structure directly from data, eliminating reliance on heuristic rules or domain-specific design. Unlike prior graph structure learning methods that jointly optimize the graph and its downstream graph classifier, our method operates independently from the graph encoder, enabling flexible and architecture-agnostic graph generation.

- A variance-aware statistical sparsification mechanism: We introduce mean- and max-bound filtering strategies that convert dense and noisy attention patterns into sparse, reliable document graphs by selecting statistically salient inter-sentence dependencies.

- Strong performance gains over heuristic graphs and non-graph baselines: Across datasets, our learned graphs improve accuracy by up to 4 points and macro-$F_1$ by 4.3 points over heuristic graphs, and by up to 2.7 points over recent small language models.

- Comprehensive evaluation and analysis: We benchmark our approach against five heuristic-based graph construction methods, two transformer baselines, and two embedding-based baselines, and analyze structural properties, resource usage, and the behavior of our filtering strategies on three publicly available datasets.

## 2 RELATED WORK

### 2.1 PREDEFINED GRAPH SCHEMES

**Classic Approaches.** Numerous graph-based text representation approaches have been used in text classification, demonstrating their effectiveness in capturing textual relationships. Early methods primarily relied on co-occurrence statistics and linguistic patterns, often representing words as nodes and connecting them if they co-occur within a fixed-size sliding window (Mihalcea & Tarau, 2004; Hassan & Banea, 2006; Rousseau et al., 2015; Zhang et al., 2020). Sequence-based graphs offer a complementary approach, connecting words based on their original order. While early implementations used weighted edges by frequency (Castillo et al., 2015), recent work suggests that binarized edges can improve performance (Bugueño & de Melo, 2023).

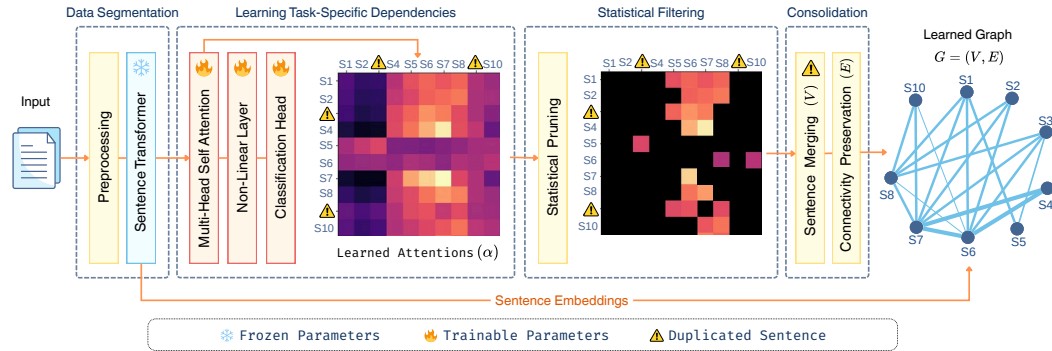

Figure 2: Overview of our framework. Non-trainable steps include *data segmentation*, *statistical filtering*, and *consolidation*, while the Sentence Transformer is used with frozen parameters. Edge widths in graph $G$ reflect learned edge weights, with thicker edges denoting stronger dependencies.

**Recent Approaches.** More sophisticated methods have been introduced to enhance textual modeling. TextGCN (Yao et al., 2019) builds a global heterogeneous graph with word and document nodes, using Point-wise Mutual Information (PMI) for weighting word–word edges and TF-IDF for word–document links, whereas TextLevelGCN (Huang et al., 2019) generates one graph per document by connecting word nodes (duplicated when repeated) based on co-occurrence within a sliding window, also weighted by PMI.

Extensions enrich graphs with additional heterogeneous elements, such as topic nodes (Gu et al., 2023; Cui et al., 2020), word and character n-grams (Li & Aletras, 2022), label nodes (Li et al., 2024), or document metadata including keywords, entities, and titles (Ai et al., 2023). Other studies incorporate multiple edge types while maintaining a single node type, encoding features such as titles, keywords, or events for document nodes (Ai et al., 2025), or combining co-occurrence, syntactic dependencies, and self-loops for word nodes (Wang et al., 2023). An alternative strategy (Li et al., 2025b) constructs separate graphs for words and sentences that are fused during training.

**Limitation.** Despite their progress, existing methods share a fundamental limitation: reliance on predefined domain knowledge to establish node and edge types, making them heavily task- and domain-specific. To address this, a learning-based approach for automatic graph structure discovery can offer a more generalizable and adaptable alternative by removing the need for manual design.

## 2.2 LEARNING THE DOCUMENT STRUCTURE

To the best of our knowledge, no existing method learns document graph structures directly from raw text. Instead, they depend on domain-specific heuristics for graph construction. While some studies incorporate graph-based techniques to enrich contextual representations, they do not explicitly learn the graph topology itself.

The most relevant work (Xu et al., 2021) combines a Graph Attention Network (GAT) (Veličković et al., 2017) with a pre-trained Transformer encoder. In this approach, documents are segmented into passages, encoded using RoBERTa (Liu et al., 2019), and structured as fully connected subgraphs linked to a central node representing the document. A GAT captures multi-granularity document representations, while contrastive learning further enhances representation learning. While effective, this method does not learn the underlying graph topology. Rather, it relies on a fixed architecture and predefined connections.

Recent research explores integrating predefined static graph structures with language models to enhance document representation learning by combining local relational modeling with deep contextual encoding. One such study (Huang et al., 2022) proposes a unified model combining Graph Neural Network (GNN) models and BERT (Devlin et al., 2019) to mitigate the overemphasis on content-specific word usages. The method employs a sub-word graph to learn fine-grained syntactic relationships. Similarly, Onan (2023) introduce a hierarchical graph-based framework where BERT encodes contextual information at the node level, enhancing classification performance.

In multi-label classification, Liu et al. (2025) use XLNet (Yang et al., 2019) embeddings and generate a graph structure based on label co-occurrence, learning label correlations exclusively through graph attention. Moreover, class-specific and self-attention mechanisms enhance the model's ability to capture contextual dependencies within the text.

**Graph Structure Learning (GSL).** GSL (Franceschi et al., 2019; Chen et al., 2020) emerged as an alternative to heuristic-based graph constructions, whose manually specified rules often yield incomplete or task-misaligned topologies. GSL addresses these limitations by enabling parametric graph induction, wherein graph topology and node representations are jointly optimized.

Early approaches refine an initial graph using shallow feature embeddings and typically follow a metric-based method. Among them, IDGL (Chen et al., 2020) iteratively infers edges via node embedding similarity, while SE-GSL (Zou et al., 2023) applies an entropy-based abstraction to form hierarchical communities. More recent work leverages large language models (LLMs) to reduce reliance on explicit graph structural information as supervision signals. GraphEdit (Guo et al., 2024) uses instruction-tuning to fine-tune an LLM to predict edge relevance from node text, whereas LLaTA (Zhang et al., 2025) leverages tree-based in-context learning to integrate topology and text insights. While incorporating external knowledge improves robustness against noisy input graphs, such methods suffer from substantial computational inefficiencies, limiting their scalability to large graphs. Moreover, most GSL methods target non-textual domains such as citation and social networks (Franceschi et al., 2019; Chen et al., 2020; Zou et al., 2023). When applied to text, they generally operate on corpus-level graphs, treating each document as a node. This formulation neglects the semantic dependencies across text granularities, thereby restricting applicability to short or structurally simple documents. Moreover, graph structure learning in these cases is jointly optimized with the underlying graph encoder, thereby coupling the learned structure with model-specific inductive biases.

Despite the growing use of graph-based methods, the challenge of automatically learning graph topologies for document representation directly from raw text remains largely unexplored. Recent efforts underscore the effectiveness of integrating attention mechanisms and pre-trained language models for robust and adaptive graph-based document representations, while also highlighting the limitations of heuristic-based graph constructions–particularly in handling diverse domains and coping with modern document processing requirements such as large-scale data, long-range dependencies, and noisy and imbalanced data.

## 3 LEARNING DATA-DRIVEN DOCUMENT GRAPHS

We introduce a novel approach for learning data-driven graph structures, eliminating the reliance on manual graph design and minimizing domain dependency. Building upon insights from previous work (Xu et al., 2021; Liu et al., 2025) highlighting the capabilities of pre-trained language models and attention mechanisms for capturing contextual relationships (Voita et al., 2019; Clark et al., 2019), our methodology constructs homogeneous weighted graphs, where sentences serve as nodes and inter-sentence dependencies are learned via a self-attention mechanism.

**Motivation for Attention-Based Graphs.** Self-attention computes pairwise relevance scores between representational units conditioned on task supervision. Let $D$ be a document with $L$ sentences $s_1, s_2, \ldots, s_L$, $X \in R^{L \times d}$ the sentence embedding matrix, and $\alpha = \text{Attn}(X) \in R^{L \times L}$ the resulting sentence-level attention matrix (aggregated across heads). Under mild assumptions, large entries $\alpha_{ij}$ indicate that $s_j$ strongly influences $s_i$'s contribution to the document-level prediction. Thus, thresholding $\alpha$ yields a data-adaptive sparsified graph $G = (V, E)$, where $V = \{s_i, s_2, \ldots, s_L\}$ and $E = \{(s_i, s_j) \mid \alpha_{ij} \geq \tau_i\}$, with each edge weighted by its corresponding attention score $\alpha_{ij}$ and $\tau_i$ denoting a pre-calculated attention threshold for every sentence $s_i \in D$. This allows the graph to capture functional rather than purely surface interactions, resulting in a lightweight alternative to domain-engineered topologies. Notably, the resulting topology adapts to each document, enabling different sparsity patterns even among texts with similar surface characteristics.

Our use of sentences as nodes is motivated by their proven effectiveness in capturing document structure and their scalability for long texts (Song et al., 2020). Furthermore, we generate homogeneous rather than heterogeneous graphs to avoid the computational overhead and reliance on external

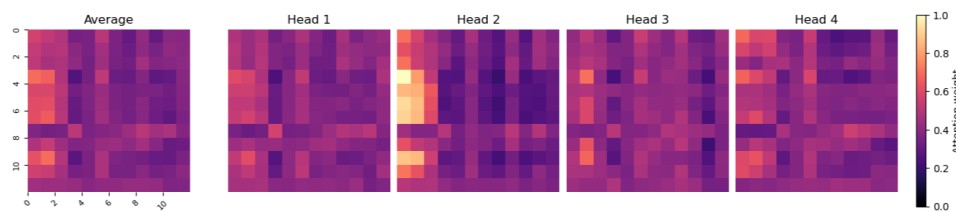

Figure 3: Learned attention weights averaged across heads for a randomly selected document from the BBC News dataset, consisting of 12 sentences.

tools (Sahu et al., 2019; Wang et al., 2023; Ai et al., 2025). Previous work also suggests that simpler graph constructions often perform better than more specialized ones (Bugueño & de Melo, 2023).

After learning the attention weights for all sentence pairs, a statistical filtering mechanism defines a minimum threshold for each row $i$ ($\tau_i$) in the attention matrix, ensuring that only strongly correlated sentence pairs ($\alpha_{ij}$) are retained. This step also mitigates isolated nodes, enhancing the graph quality and reducing graph complexity. The overall framework is outlined in Figure 2, with a detailed description provided below.

### 3.1 DATA SEGMENTATION

To define the units that will serve as nodes within the learned graphs, namely, the sentences, our approach begins by segmenting the input document $D$ into a sequence of its $L$ preprocessed constituent sentences $s_1, s_2, \ldots, s_L$. For this, we conduct a data-cleaning procedure followed by sentence tokenization.[2] Sentences containing fewer than five words are merged with the preceding one to prevent the graph size from growing excessively, ensure computational efficiency, and maintain meaningful sentence representations while reducing unnecessary complexity in graph construction. This segmentation allows the model to later capture sentence-level dependencies that are essential to accurately modeling the overall structure of the document graph.

### 3.2 LEARNING TASK-SPECIFIC DEPENDENCIES

Each sentence $s_i$ is embedded into a fixed-dimensional vector $x_i \in \mathrm{R}^d$ using a pre-trained Sentence Transformer, with $d = 384$ in our experiments. The resulting set of embeddings $x_i, x_2, \ldots, x_L$ serves as the input representation of the document, effectively transforming the textual data into vector representations for further processing.

Building upon these representations, a multi-head self-attention model is trained to learn inter-sentence dependencies. The architecture comprises a multi-head attention mechanism, followed by a ReLU-activated non-linear layer, and concludes with a classification head designed to perform document classification across the available classes. Following recent findings (Wortsman et al., 2023), we substitute the conventional softmax activation function used during the scaled dot-product attention computation with a ReLU activation normalized by sequence length. This variant has been shown to yield a more efficient and effective attention mechanism (Bai et al., 2023; Zhao et al., 2024). The resulting learned attention matrix is given by $\alpha_{ij}$ for each sentence pair $(s_i, s_j)$. Motivated by prior work demonstrating that self-attention heads often correspond to interpretable linguistic patterns (Voita et al., 2019; Clark et al., 2019), we compute $\alpha_{ij}$ by averaging the attention matrices across all heads in the model (see Figure 3). Additional implementation details and visualizations of the resulting attention distributions are provided in Appendix B.

### 3.3 STATISTICAL FILTERING

To enhance the relevance of the attention weights produced by the multi-head self-attention model, we apply a statistical filtering step that selectively discards weak dependencies while retaining salient sentence pairs ($\alpha_{ij}$) relevant to the classification task. This process effectively transforms attention weights into graph edges representing meaningful inter-sentence relationships. Filtering

---

[2]Implemented using the NLTK library in Python.

is conducted row-wise to prevent isolated nodes, establishing at least one edge per sentence, while self-loops are explicitly removed. Two alternative filtering strategies are introduced.

**Mean-bound.** This approach computes the average attention score for each sentence $s_i$ across all other sentences within the document and derives a minimum attention threshold incorporating a predefined tolerance degree $\delta$. The threshold is given by:

$$\tau_i = \frac{1}{L} \sum_{j=1}^{l} \alpha_{ij} + \delta \cdot \text{std}(\alpha_i) \,, \tag{1}$$

where $\text{std}(\alpha_i)$ is the standard deviation of the $i$-row of the learned attention matrix. This threshold is slightly greater than the mean, which reduces the tolerance level and decreases the number of retained entries in the attention matrix, thereby ensuring that only the most relevant dependencies are preserved.

**Max-bound.** This strategy focuses on top-ranked dependencies, retaining attention scores proximate to the maximum observed value within each row, i.e., for each sentence $s_i$ in the document. The threshold is calculated as:

$$\tau_i = \max_j(\alpha_{ij}) - \delta \cdot \text{std}(\alpha_i) \,, \tag{2}$$

where $\text{std}(\alpha_i)$, as in Equation 1, is the standard deviation of the $i$-row of the learned attention matrix. Notably, we increase the tolerance for preserving entries around the peak attention score for each row, yielding a more aggressive pruning criterion.

### 3.4 Consolidation

After statistical filtering, the resulting matrix is treated as the adjacency matrix of the learned graph. To ensure structural consistency, two operations account for special edge cases.

**Sentence Merging.** When identical sentences occur at different positions in $D$, their corresponding edges are merged to maintain the integrity of the graph representation and better reflect the semantic structure of the document, while adjusting the set of effective sentence nodes in the final learned graph. For instance, in Figure 2, $s_3 = s_9$ in $D = \{s_1, s_2, \cdots, s_{10}\}$. Therefore, their edges are unified, resulting in a reduced graph with nine unique sentence nodes.

**Connectivity Preservation.** Isolated nodes need to be avoided. A typical scenario arises when there is no plausible edge for the row $\alpha_i$ ($s_i$) after statistical filtering, failing to establish meaningful connections. To address this, additional edges are introduced between $s_i$ and its immediately preceding and subsequent sentences. The attention weight associated with the self-loop $\alpha_{ii}$ is evenly distributed between these new edges, preserving the original attention-based weighting scheme.

The final learned graph $G = (V, E)$ consists of unique sentence nodes $V \subseteq D$, encoded via Sentence Transformer embeddings, and undirected, attention-weighted edges $E$ that effectively capture the document structure. More details about the graph induction from $\alpha$ are available in Appendix D.

## 4 Experiments

To study the merits of our learned graphs for document representation in various scenarios, we conducted experiments on three publicly available text classification datasets, comparing our approach against five heuristic-based construction schemes by training a GAT under consistent settings.

### 4.1 Datasets

We assess our model's generalizability across balanced and unbalanced scenarios, covering topic classification and hyperpartisan news detection (see Table 1), using datasets of varying length: **BBC News**[3] (Greene & Cunningham, 2006), a moderately imbalanced collection of English news articles

---

[3] http://derekgreene.com/bbc/

Table 1: Statistics of datasets. Imbalance rate (IR) denotes the ratio of minority to majority classes.

| Dataset | #Samples (train/val/test) | Avg. Length | #Classes | IR |
|---|---|---|---|---|
| BBC News | 2,225 (1,547/177/443) | 438 words (19 sent.) | 5 | 4:5 |
| HND | 1,270 (516/129/625) | 912 words (21 sent.) | 2 | 1:2 |
| arXiv | 33,000 (28,000/2,500/2,500) | 10,554 words (539 sent.) | 11 | 1:2 |

in the areas of business, entertainment, politics, sport, and technology; **Hyperpartisan News Detection (HND)**[4] (Kiesel et al., 2019) binary annotated for partisan bias; and **arXiv**[5] (He et al., 2019), a corpus of 33,000 scientific papers in physics, mathematics, computer science, and biology.

As BBC News lacks predefined data splits, we apply an 80/20 train–test partition, allocating 10% of the training for validation. Duplicate entries are removed across all datasets. Further dataset details are provided in Appendix A.

## 4.2 COMPARISON METHODS

**Heuristic-based Baselines.** We compare our learned graphs against five widely adopted heuristic-based homogeneous graph constructions. While recent work also explores heterogeneous graphs (Section 2.1), they differ fundamentally from our homogeneous setup and are not directly comparable. In all baselines, graph nodes represent the unique sentences in a document $D$. We consider:

- **Complete Graph**: A fundamental baseline, where each sentence node is fully connected to all others using unweighted edges, forming a complete graph.

- **Sentence Order**: Constructs undirected binary edges based on the natural order of sentence occurrence within the document. This approach solely captures the sequential structure of the text.

- **Window-based Co-Occurrence**: Undirected edges connect sentence nodes if they co-occur within a fixed sliding window of size 3. Therefore, each sentence is connected to its two preceding and two subsequent sentences. Notably, this construction can be considered a generalization of the sentence order-based graph by capturing broader contextual dependencies.

- **Mean Semantic Similarity**: Defines weighted edges based on a cosine similarity threshold applied to the corresponding sentence embeddings. The threshold is computed as described in Equation 1, providing a fair comparison against our learned graphs.

- **Max Semantic Similarity**: Sets a higher cosine similarity threshold by following the procedure outlined in Equation 2. It retains only the strongest connections, resulting in sparser graphs.

**Non-Graph Baselines.** We include strong non-graph baselines in Table 2 for comparative reference. We fine-tune **Longformer** (Beltagy et al., 2020) using the pretrained `longformer-base-4096`[6] model with a sequence-classification head. **LongT5** (Guo et al., 2022) is similarly fine-tuned using only the encoder of the pre-trained `google/long-t5-tglobal-base`[7] model, augmented with a linear classification head. We additionally report previously published results for each dataset, including: fine-tuned **RoBERTa** (Reusens et al., 2024; Liu et al., 2019) with Bayesian-optimized hyperparameters for BBC News; **CogLTX** (Park et al., 2022; Ding et al., 2020), which selects informative sentences for document classification, for HND; and **Llama3.2** (Li et al., 2025a; Touvron et al., 2023) for the arXiv dataset. An extensive comparison including additional non-graph models is provided in Table 6. We further implement a simple baseline that represents each document as the mean of its Sentence Transformer embeddings, followed by a two-layer MLP (**SentenceEmb + MLP**). This model operates entirely independently of graph structure and provides a controlled lower bound against which the gains of our graph-based architectures can be assessed. We also report results from our sentence-level self-attention models (**ReLUAttn-SentenceEmb + MLP**), which induce our learned graphs, as described in Section 3.2.

---

[4] https://zenodo.org/records/5776081

[5] https://huggingface.co/datasets/ccdv/arxiv-classification

[6] https://huggingface.co/allenai/longformer-base-4096

[7] https://huggingface.co/google/long-t5-tglobal-base

Table 2: Classification performance of learned versus heuristic-based and non-graph models across datasets. Metrics include accuracy, macro-averaged $F_1$ score (mean $\pm$ std. over 5 independent runs), graph statistics (average number of nodes, edges, and degree), and storage. Results for complete and mean semantic similarity baselines are excluded on arXiv due to excessive computational and runtime requirements. **Best**, second-best, and non-graph[†] results are highlighted as described. N/A entries indicate that the corresponding feature is not applicable (i.e., for non-graph models).

| Scheme | Accuracy | $F_1$-ma | \|V\| | \|E\| | Degree | Disk |
|---|---|---|---|---|---|---|
| *BBC News (2L-64U)* | | | | | | |
| RoBERTa (Reusens et al., 2024)[†] | 98.0 | 97.0 | N/A | N/A | N/A | N/A |
| Longformer-base[†] | 97.9 $\pm$ 0.5 | 97.8 $\pm$ 0.5 | N/A | N/A | N/A | N/A |
| LongT5-tglobal-base[†] | 96.3 $\pm$ 0.5 | 96.3 $\pm$ 0.5 | N/A | N/A | N/A | N/A |
| SentenceEmb + MLP[†] | 95.2 $\pm$ 0.3 | 94.8 $\pm$ 0.3 | N/A | N/A | N/A | N/A |
| ReLUAttn-SentenceEmb + MLP[†] | 95.9 $\pm$ 0.4 | 95.7 $\pm$ 0.4 | N/A | N/A | N/A | N/A |
| Complete Graph | **99.9 $\pm$ 0.1** | **99.9 $\pm$ 0.1** | 19.30 | 481.67 | 18.3 | 105MB |
| Sentence Order | 99.7 $\pm$ 0.3 | 99.7 $\pm$ 0.4 | 19.30 | 36.61 | 1.87 | 74MB |
| Window Co-occurrence | 99.8 $\pm$ 0.3 | 99.8 $\pm$ 0.4 | 19.30 | 71.21 | 3.62 | 76MB |
| Mean Semantic Similarity | 99.4 $\pm$ 0.5 | 99.3 $\pm$ 0.6 | 19.30 | 159.68 | 5.40 | 84MB |
| Max Semantic Similarity | 99.7 $\pm$ 0.2 | 99.7 $\pm$ 0.2 | 19.30 | 36.66 | 1.88 | 74MB |
| Learned Mean-Bound | **99.9 $\pm$ 0.1** | **99.9 $\pm$ 0.1** | 19.30 | 213.80 | 8.26 | 90MB |
| Learned Max-Bound | 99.6 $\pm$ 0.5 | 99.6 $\pm$ 0.5 | 19.30 | 60.27 | 3.04 | 77MB |
| *HND (3L-64U)* | | | | | | |
| CogLTX (Park et al., 2022)[†] | 94.8 | Not Reported | N/A | N/A | N/A | N/A |
| Longformer-base[†] | 85.4 $\pm$ 1.0 | 85.3 $\pm$ 1.0 | N/A | N/A | N/A | N/A |
| LongT5-tglobal-base[†] | 74.6 $\pm$ 1.7 | 74.5 $\pm$ 1.7 | N/A | N/A | N/A | N/A |
| SentenceEmb + MLP[†] | 74.8 $\pm$ 1.0 | 74.7 $\pm$ 1.0 | N/A | N/A | N/A | N/A |
| ReLUAttn-SentenceEmb + MLP[†] | 75.4 $\pm$ 0.7 | 75.3 $\pm$ 0.7 | N/A | N/A | N/A | N/A |
| Complete Graph | 94.6 $\pm$ 1.2 | 94.5 $\pm$ 1.2 | 19.48 | 710.55 | 18.48 | 70MB |
| Sentence Order | 92.6 $\pm$ 2.3 | 92.6 $\pm$ 2.3 | 19.48 | 36.99 | 1.79 | 43MB |
| Window Co-occurrence | 92.1 $\pm$ 2.9 | 92.1 $\pm$ 2.9 | 19.48 | 71.94 | 3.35 | 44MB |
| Mean Semantic Similarity | 91.2 $\pm$ 4.9 | 91.1 $\pm$ 5.0 | 19.48 | 253.77 | 6.00 | 53MB |
| Max Semantic Similarity | 92.8 $\pm$ 5.5 | 92.8 $\pm$ 5.6 | 19.48 | 36.93 | 1.78 | 43MB |
| Learned Mean-Bound | 95.0 $\pm$ 2.2 | **94.9 $\pm$ 2.2** | 19.48 | 293.25 | 7.85 | 56MB |
| Learned Max-Bound | 92.6 $\pm$ 5.6 | 92.6 $\pm$ 5.6 | 19.48 | 54.76 | 2.53 | 44MB |
| *arXiv (3L-64U)* | | | | | | |
| Llama-3.2-1B-Instruct (Li et al., 2025a)[†] | 89.2 | 89.0 | N/A | N/A | N/A | N/A |
| Longformer-base[†] | 86.9 $\pm$ 0.8 | 86.7 $\pm$ 0.7 | N/A | N/A | N/A | N/A |
| LongT5-tglobal-base[†] | 87.8 $\pm$ 0.7 | 87.8 $\pm$ 0.7 | N/A | N/A | N/A | N/A |
| SentenceEmb + MLP[†] | 79.9 $\pm$ 0.3 | 79.1 $\pm$ 0.3 | N/A | N/A | N/A | N/A |
| ReLUAttn-SentenceEmb + MLP[†] | 84.7 $\pm$ 0.6 | 84.1 $\pm$ 0.6 | N/A | N/A | N/A | N/A |
| Sentence Order | 87.8 $\pm$ 0.5 | 87.3 $\pm$ 0.5 | 510.33 | 1,034.28 | 2.02 | 25GB |
| Window Co-occurrence | 87.9 $\pm$ 1.9 | 87.4 $\pm$ 2.0 | 510.33 | 2,064.93 | 4.03 | 26GB |
| Max Semantic Similarity | 87.8 $\pm$ 0.8 | 87.3 $\pm$ 0.8 | 510.33 | 1,234.40 | 2.27 | 26GB |
| Learned Mean-Bound (10% train sample) | 88.2 $\pm$ 3.1 | 87.6 $\pm$ 3.2 | 502.92 | 35,651.40 | 55.53 | 16GB |
| Learned Max-Bound (10% train sample) | 91.7 $\pm$ 1.9 | 91.3 $\pm$ 2.1 | 502.92 | 1068.51 | 2.14 | 6GB |
| Learned Max-Bound (full data) | **91.9 $\pm$ 1.1** | **91.7 $\pm$ 1.0** | 510.33 | 1082.22 | 2.14 | 25GB |

### 4.3 EXPERIMENTAL SETUP

We established dataset-specific maximum sequence lengths to handle particularly long documents. While BBC News and HND retained full texts, a maximum of 1,800 sentences was set for arXiv, minimizing information loss and truncating less than 1.5% of document samples. To obtain sentence embeddings, we used the pre-trained Sentence Transformer `paraphrase-MiniLM-L6-v2`[8]. The tolerance degree $\delta$ is fixed at 0.5 throughout all experiments.

**Self-Attention Model:** We employed a single-layer four-head multi-head self-attention model, trained with a batch size of 32 for up to 20 epochs; however, additional experiments with a two-layer architecture are reported in Table 4, Appendix B. Optimization was performed using Adam (Kingma & Ba, 2014) with an initial learning rate of 0.001, employing early stopping if the validation macro-averaged $F_1$ score did not improve for five consecutive epochs.

**Graph Attention Network (GAT):** We assessed GAT architectures with 1 to 3 hidden layers and node embedding sizes in $\{64, 128, 256\}$. Dropout (rate=0.2) was applied after each convolutional

---

[8]`https://huggingface.co/sentence-transformers/paraphrase-MiniLM-L6-v2`

layer, edge weights were used as edge attributes, and average pooling aggregated node embeddings. Final document representations were classified through a softmax layer. Training was conducted for up to 50 epochs with a batch size of 64, utilizing the Adam optimizer (Kingma & Ba, 2014) and an initial learning rate of 0.001. Early stopping was applied as in self-attention models.

**Language Models:** Longformer and LongT5 are trained with a learning rate of $2 \times 10^{-5}$, a maximum of 10 epochs, and patience of 3. Due to the model size (148M and 109M parameters, respectively) and computational constraints, we set the maximum sequence lengths to 1024, 2048, and 4096 for each dataset, while LongT5 uses a maximum length of 5120 for arXiv. Batch size is fixed to 16.

## 5 RESULTS

Table 2 reports main results averaged over 5 runs, with accuracy and macro-averaged $F_1$ score accounting for class imbalance. Given the substantially longer documents in arXiv, GATs were first trained on 10% of the training samples for the learned graph variants (mean- and max-bound), and the best-performing model was subsequently scaled to the full dataset. GAT architectures were adapted to the dataset length: 2-layer GAT (64 units) for BBC News, and 3-layer GAT (64 units) for HND and arXiv, capturing the complex semantic relationships present in lengthy documents. Importantly, additional experiments using two alternative GNN backbones–Graph Convolutional Networks (GCN) (Kipf & Welling, 2017) and GraphSAGE (Hamilton et al., 2017)–are provided in Table 5; Appendix C.
All experiments were implemented in PyTorch Geometric on an NVIDIA GeForce RTX3050.

**Quality of the Results.** Learned graphs consistently outperform heuristic-based schemes across datasets, showing robust performance and proving competitive with non-graph methods, surpassing recent small language models. Gains over heuristic graphs are marginal on BBC News but become increasingly pronounced for longer documents. Notably, although the complete graph baseline matches the performance of our learned mean-bound graphs, it requires nearly twice the edges and 15 MB more storage. On HND, learned mean-bound graphs outperform the strongest heuristic baseline (max semantic similarity) by up to 2.1 $F_1$ points, with even larger improvements on arXiv (4.3 $F_1$ over window-based graphs, 2.7 $F_1$ over Llama-3.2). Remarkably, training on only 10% of the training data, our learned graphs outperform heuristic schemes and Longformer-base, with the max-bound variant surpassing Llama-3.2. These results emphasize the effectiveness of our method in capturing structural information in long texts.

Moreover, although ReLUAttn-SentenceEmb + MLP shows improvements over SentenceEmb + MLP, both methods perform well on mid-length documents but degrade substantially as length increases, highlighting the limitations of flat sentence pooling. In contrast, our learned graphs achieve consistent gains–reaching up to +20 $F_1$ on HND and +7 $F_1$ on arXiv. These findings indicate that modeling document structure, rather than relying solely on sentence-level content, is essential for long-text classification.

**Graph Structure Analysis.** A key advantage of our proposal is its ability to capture global contextual dependencies within a document. Unlike heuristic graphs, which are limited to local context via fixed window sizes, our approach allows edges between distant but relevant sentences, considering all sentences simultaneously and thereby enhancing the expressiveness of the learned structure. Despite comparable storage requirements and average degree, our learned mean-bound graphs substantially outperform heuristic-based mean semantic similarity graphs. This indicates that the performance gains stem not from graph density but from the semantic relevance and structural alignment of the learned edges. On arXiv, even the strongest heuristic baselines (window co-occurrence and max semantic similarity graphs) exhibit higher average degrees than our learned max-bound graphs, yet achieve lower performance, underscoring the robustness and effectiveness of our approach.

Visualizations of adjacency matrices (Figure 4) underscore the importance of capturing comprehensive document structures, highlighting the significance of both initial and final sentences for accurate classification, particularly in long-form documents.For clarity, we include binarized versions of the learned adjacency matrices, as they typically exhibit lower edge weights than heuristic-based graphs.

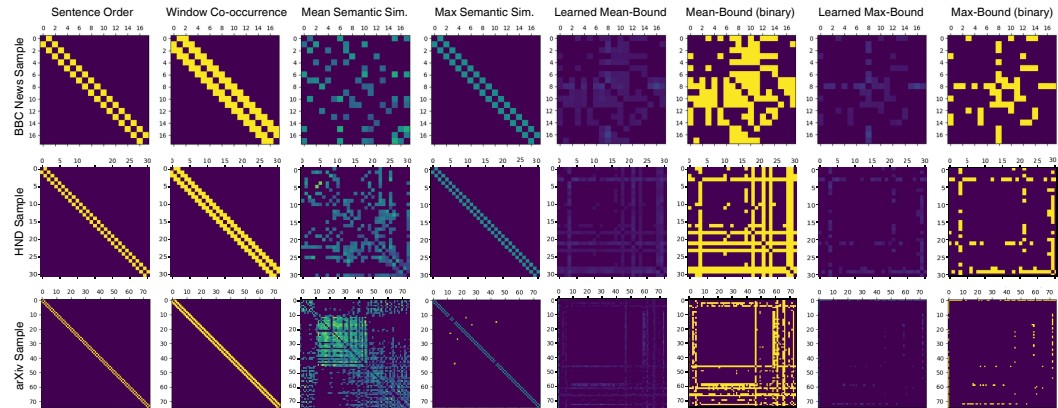

Figure 4: Adjacency matrix comparison across graph schemes on random dataset samples.

Table 3: Ablation results on the HND dataset.

| $\delta$ | Mean-bound | | | Max-bound | | |
|---|---|---|---|---|---|---|
| | $F_1$-ma | Degree | Disk | $F_1$-ma | Degree | Disk |
| 0.25 | 93.4 | 11.47 | 61MB | 94.0 | 2.40 | 44MB |
| 0.50 | **94.9** | 8.86 | 57MB | 92.6 | 2.79 | 45MB |
| 0.75 | 94.8 | 6.78 | 53MB | 91.6 | 3.29 | 45MB |
| 1.00 | 90.7 | 5.20 | 50MB | 94.2 | 3.90 | 46MB |
| – | No Filter: $F_1$-ma: 89.6 / Degree: 39.96 / Disk: 104MB | | | | | |

**Ablation and Sensitivity Analysis.** Table 3 reports the contribution of key design choices on the HND dataset: tolerance degree ($\delta$), and filtering strategy. For mean-bound filtering, optimal performance is achieved with $\delta$ values between 0.5 and 0.75, with macro-averaged $F_1$ scores near 95%. In contrast, retaining all edges close to the row-wise mean attention score results in overly dense graphs (11.47 neighbors on average), leading to semantically noisy and undifferentiated message passing in the GAT, which degrades classification accuracy. This issue is more notorious when no filtering is applied and the full attention matrix is used as the adjacency matrix: The average node degree rises to nearly 40, and the $F_1$ score drops to 89.6%. Conversely, increasing the threshold to 1.0 substantially reduces the number of edges, proving insufficient for the task.

For the max-bound strategy, the GAT achieves competitive results by retaining only those edges close to the row-wise maximum attention value, with a considerable decay when increasing the tolerance degree. Notably, higher tolerance values in this setting retain more edges, in contrast to mean-bound, where higher thresholds produce sparser graphs. Interestingly, at $\delta = 1.0$, performance partially recovers, resembling the results obtained by mean-bound filtering with a higher average node degree.

These findings underscore the importance of statistical filtering to maintain a balance between graph sparsity and semantic relevance. They also suggest a need for further investigation into the interplay between edge semantics, node degree, and downstream task performance.

# 6 CONCLUSION

We introduced a data-driven framework that induces document graphs from supervised self-attention and prunes them via statistical filtering, eliminating reliance on domain-specific heuristics. Comprehensive experiments on three document classification datasets demonstrate that our learned graphs consistently outperform strong heuristic-based baselines, capturing the long-range and non-sequential dependencies that sentences may have among themselves. An ablation and sensitivity analysis confirms the importance of attention-guided sparsification and connectivity preservation. Future work includes adaptive threshold learning, additional tasks, and hierarchical extensions that incorporate multi-granular textual structure.

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

## A  DATASETS

**BBC News**[9] (Greene & Cunningham, 2006): A moderately imbalanced collection of 2,225 English documents from the BBC News website (2004–2005) in the areas of business, entertainment, politics, sport, and technology. As BBC News lacks predefined data splits, after duplicate removal, we partition the data into training (1,547), validation (177), and test (443) sets. Notably, the dataset is available for non-commercial and research purposes only.

**Hyperpartisan News Detection (HND)**[10] (Kiesel et al., 2019): English news articles labeled according to whether they show blind or unreasoned allegiance to a single political party or entity, or not. Although it comprises two parts, `byarticle` and `bypublisher`, we use the first one with 645 training and 625 test samples labeled through crowdsourcing. As HND does not have a predefined validation split, we reserve 10% of the training samples for such a purpose. The collection is licensed under a Creative Commons Attribution 4.0 International License.

**arXiv**[11] (He et al., 2019): A collection of 33,000 very long scientific papers in physics, mathematics, computer science, and biology sourced from arXiv. The documents were originally obtained in PDF format and subsequently converted into plain text using the arXiv sanity preserver tool[12]. The corpus

---

[9] http://derekgreene.com/bbc/
[10] https://zenodo.org/records/5776081
[11] https://huggingface.co/datasets/ccdv/arxiv-classification
[12] https://github.com/karpathy/arxiv-sanity-preserver/

is organized into 11 classes with a slight class imbalance, and partitioned into three splits: train (28,000), validation (2,500), and test (2,500).

# B  SELF-ATTENTION FORMULATION

## B.1  ReLU-BASED ATTENTION

Self-attention provides a mechanism for modeling long-range dependencies by enabling each token in a sequence to attend to all others, thus producing non-local contextualized representations. This is achieved by projecting the input into three parameterized spaces (queries $(Q)$, keys $(K)$, and values $(V)$) which define pairwise interactions across tokens. Classical self-attention computes token-level similarity scores via scaled dot-products between queries and keys, followed by a nonlinear transformation (typically softmax) that yields normalized attention weights. These weights are then used to aggregate information from the values, producing context-aware token embeddings. Formally, a single-head self-attention mechanism is often written as:

$$\alpha = \text{Attn} = \phi\left(\frac{QK^T}{\sqrt{d_k}}\right)V, \tag{3}$$

where $\phi(\cdot)$ denotes the nonlinear activation (softmax in the standard formulation), $d_k = \frac{d}{H}$ is the head dimension, and $H$ heads are used in parallel to form multi-head self-attention.

In our setting, let be $X \in \mathbb{R}^{L \times d}$ denote the input representation of a document with at most $L$ sentence-tokens, each embedded into a $d$-dimensional space (e.g., $d = 384$ using the Sentence Transformer).

We obtain queries, keys, and values via linear projections of $X$:

$$q, k, v = XW_{q,k,v,} + b_{q,k,v} \quad \in \mathbb{R}^{L \times d}, \tag{4}$$

where $W_{q,k,v,} \in \mathbb{R}^{d \times d}$ and $b_{q,k,v} \in \mathbb{R}^d$.

These are then reshaped and partitioned into per-head matrices $Q, K, V \in \mathbb{R}^{H \times L \times d_k}$, so that for each head $h \in \{1, \dots, H\}$ we have:

$$Q^h, K^h, V^h \quad \in \mathbb{R}^{L \times d_k}. \tag{5}$$

Afterwards, we compute the scaled-dot product attention logits for each head:

$$S^h = \frac{1}{\sqrt{d_k}} Q^h (K^h)^T \quad \in \mathbb{R}^{L \times L}. \tag{6}$$

Unlike the softmax formulation, we employ the element-wise ReLU activation as the non-linearity, which is normalized by sequence length $L$:

$$\alpha_{ij}^h = \frac{\text{ReLU}(S_{ij}^h)}{L}, \quad i, j \in \{1, \dots, L\}, \tag{7}$$

yielding a non-negative but not necessarily stochastic attention distribution.

Once the attention logits are obtained, the final head-averaged attention map is given by:

$$\alpha = \frac{1}{H} \sum_{h=1}^{H} \alpha^h \quad \in \mathbb{R}^{L \times L}. \tag{8}$$

Head-specific outputs are computed as weighted value combinations, which are then concatenated to obtain the final output projection:

$$
\begin{aligned}
O^h &= \alpha^h V^h & \in \mathbb{R}^{L \times d_k}, \\
O &= \text{Concat}(O^1, \dots, O^H) & \in \mathbb{R}^{L \times d}, \\
Y &= OW_o + b_o & \in \mathbb{R}^{L \times d},
\end{aligned}
\tag{9}
$$

Table 4: Results of a 1- and a 2-layer four-head multi-head self-attention model.

| | | 1-layer | | 2-layer | |
| --- | --- | --- | --- | --- | --- |
| | | Acc | $F_1$-ma | Acc | $F_1$-ma |
| **BBC News** | sport | - | 98.7 | - | 99.1 |
| | entertainment | - | 95.2 | - | 92.9 |
| | business | - | 94.1 | - | 93.5 |
| | tech | - | 96.8 | - | 97.4 |
| | politics | - | 91.4 | - | 93.2 |
| | **macro-avg.** | 95.5 | 95.3 | 95.5 | 95.2 |
| **HND** | non-hyperpartisan | - | 75.5 | - | 76.4 |
| | hyperpartisan | - | 77.3 | - | 76.2 |
| | **macro-avg.** | 76.4 | 76.4 | 76.4 | 76.3 |

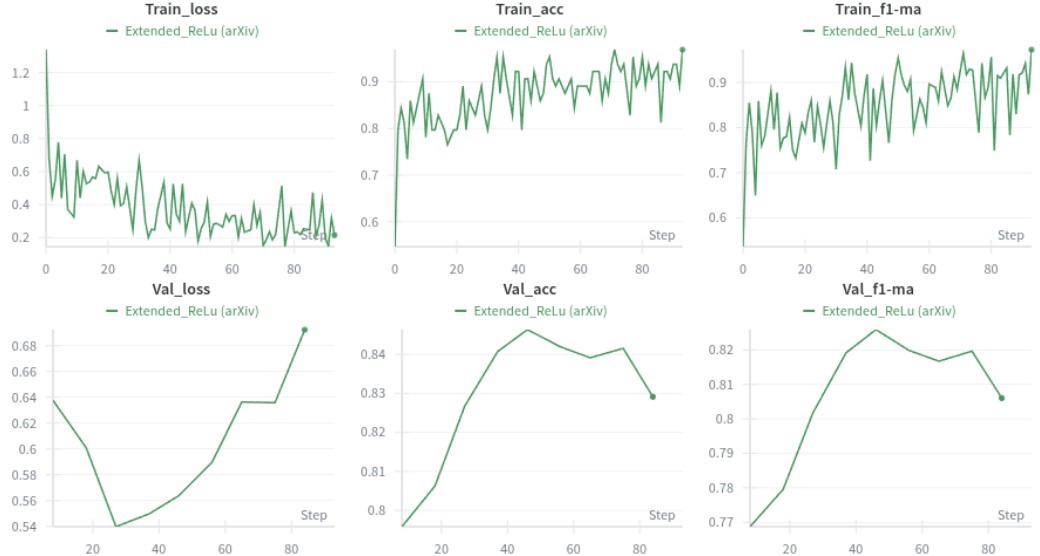

Figure 5: Learning curves of the 1-layer MHA model used for graph induction in the arXiv dataset.

where $W_o \in \mathbb{R}^{d \times d}$ and $b_o \in \mathbb{R}^d$ are learnable output projection parameters. The final document representation $Y$ thus encodes each token with information aggregated from all others through ReLU-based multi-head self-attention.

For implementation details, please refer to our public repository: `https://github.com/available/upon/publication`.

## B.2 ROBUSTNESS OF MULTI-HEAD SELF-ATTENTION MODEL

As Table 4 shows, our method demonstrates strong robustness across model architectures. Even shallow self-attention models induce strong document representations. Notably, it is essential for the learned attention weights to exhibit sparsity, which is critical for effectively identifying potential edges throughout the document. This sparsity facilitates the subsequent training of GAT models by efficiently exploring and leveraging the local neighborhood structure within the learned graph, enhancing its capacity to capture meaningful relationships within the document.

**Model Convergence**  Figure 5 presents the learning curves of the multi-head attention model used to induce graphs for the arXiv dataset. While the training loss decreases steadily throughout the optimization process, the validation loss begins to rise after the initial training steps. Early stopping is therefore applied to preserve the model checkpoint that achieves the best validation performance, as described in Section 4.3.

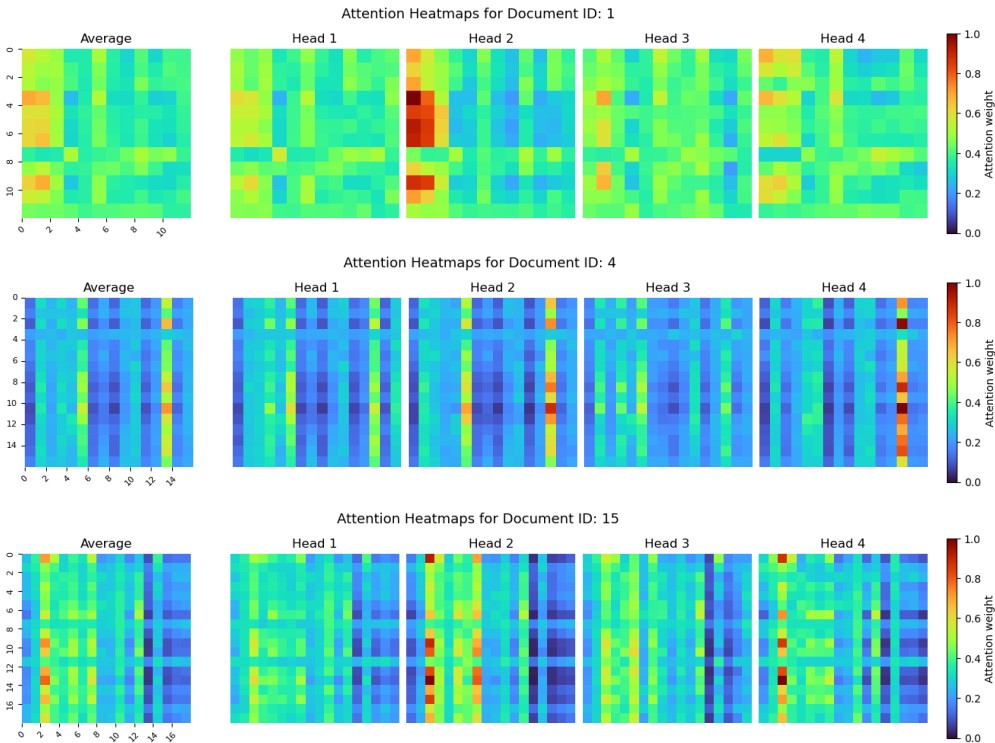

Figure 6: Learned attention weights for randomly selected samples from the BBC News dataset.

### B.3 LEARNED ATTENTION DISTRIBUTIONS

Figure 6, Figure 7, and Figure 8 visualize, as heat maps, the attention-weight distributions learned by the self-attention models used for our graph induction process across the three datasets. A different colormap is employed than in Figure 2 and Figure 3 to enhance perceptual contrast, particularly for longer documents such as those in the arXiv corpus.

As described in Section 3.2, we average the attention matrices across heads to obtain a unified representation of the model's learned dependencies. This averaging preserves patterns consistently identified as important across heads: when a head assigns a high weight to a sentence pair $(s_i, s_j)$, its contribution remains evident–albeit attenuated–in the aggregated matrix. In contrast, dependencies that are weak and detected by only a small subset of heads are further diminished through averaging, yielding values that approach zero.

## C PERFORMANCE VARIABILITY ACROSS GNN BACKBONES

We conducted additional experiments varying the type of convolutional layer used in our graph encoder models, including Graph Attention Networks (**GAT**) (Veličković et al., 2017), Graph Convolutional Networks (**GCN**) (Kipf & Welling, 2017), and **GraphSAGE** (Hamilton et al., 2017). The results are reported in Table 5. The table shows that changing the underlying graph neural network architecture yields variable performance. However, GAT consistently outperforms both GCN and GraphSAGE across the three datasets examined in this work. Furthermore, we observe that our learned graphs consistently achieve higher performance than those constructed using heuristic methods, which emphasizes that the improvements stem from the graph–construction module rather than from the choice of GNN.

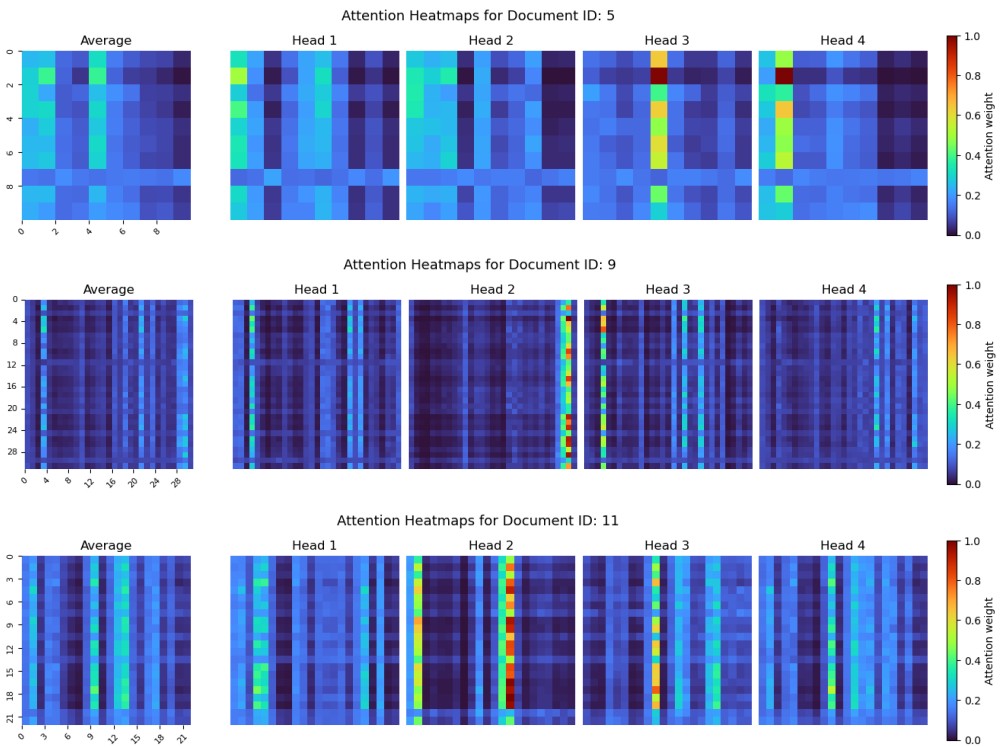

Figure 7: Learned attention weights for randomly selected samples from the HND dataset.

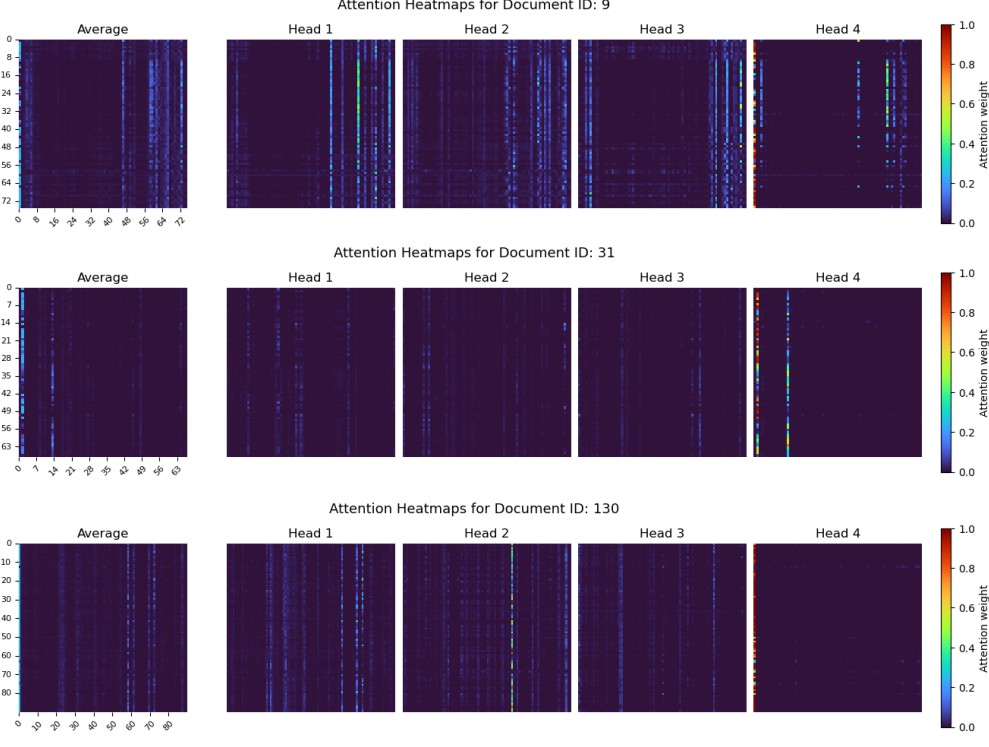

Figure 8: Learned attention weights for randomly selected samples from the arXiv dataset.

Table 5: Classification performance of learned versus heuristic-based graphs across datasets. Results encompass three different encoder models, including Graph Attention Network (GAT), Graph Convolutional Network (GCN), and GraphSAGE. Metrics include accuracy and macro-averaged $F_1$ score (mean $\pm$ std. over 5 independent runs). Results for complete and mean semantic similarity baselines are excluded on arXiv due to excessive computational and runtime requirements. **Best** and underline{second-best} results for each GNN backbone, as well as the overall $^\star$**best** result at the dataset level, are highlighted as described.

| | GAT | | GCN | | GraphSAGE | |
|---|---|---|---|---|---|---|
| **Scheme** | **Accuracy** | **$F_1$-ma** | **Accuracy** | **$F_1$-ma** | **Accuracy** | **$F_1$-ma** |
| *BBC News (2L-64U)* | | | | | | |
| Complete Graph | $^\star$**99.9 ± 0.1** | $^\star$**99.9 ± 0.1** | **99.8 ± 0.1** | **99.8 ± 0.1** | 99.5 ± 0.3 | 99.4 ± 0.3 |
| Sentence Order | 99.7 ± 0.3 | 99.7 ± 0.4 | 99.1 ± 0.4 | 99.1 ± 0.4 | **99.6 ± 0.2** | **99.6 ± 0.2** |
| Window Co-occurrence | 99.8 ± 0.3 | 99.8 ± 0.4 | 98.4 ± 1.1 | 98.3 ± 1.2 | 99.0 ± 0.5 | 98.9 ± 0.6 |
| Mean Semantic Similarity | 99.4 ± 0.5 | 99.3 ± 0.6 | 99.4 ± 0.5 | 99.4 ± 0.5 | 98.8 ± 0.6 | 98.8 ± 0.7 |
| Max Semantic Similarity | 99.7 ± 0.2 | 99.7 ± 0.2 | 99.1 ± 0.8 | 99.1 ± 0.9 | 99.4 ± 0.2 | 99.4 ± 0.2 |
| Learned Mean-Bound | $^\star$**99.9 ± 0.1** | $^\star$**99.9 ± 0.1** | 99.5 ± 0.2 | 99.4 ± 0.2 | 98.5 ± 0.7 | 98.4 ± 0.7 |
| Learned Max-Bound | 99.6 ± 0.5 | 99.6 ± 0.5 | 99.1 ± 0.3 | 99.1 ± 0.3 | 98.8 ± 0.7 | 98.8 ± 0.7 |
| *HND (3L-64U)* | | | | | | |
| Complete Graph | 94.6 ± 1.2 | 94.5 ± 1.2 | 92.3 ± 3.5 | 92.3 ± 3.5 | 91.0 ± 4.0 | 91.0 ± 4.0 |
| Sentence Order | 92.6 ± 2.3 | 92.6 ± 2.3 | 88.7 ± 3.5 | 88.7 ± 3.5 | 91.1 ± 4.0 | 91.1 ± 4.0 |
| Window Co-occurrence | 92.1 ± 2.9 | 92.1 ± 2.9 | 88.3 ± 2.4 | 88.2 ± 2.4 | 93.5 ± 0.5 | 93.5 ± 0.5 |
| Mean Semantic Similarity | 91.2 ± 4.9 | 91.1 ± 5.0 | 90.8 ± 3.8 | 90.8 ± 3.8 | 93.6 ± 1.9 | 93.6 ± 1.9 |
| Max Semantic Similarity | 92.8 ± 5.5 | 92.8 ± 5.6 | 91.8 ± 3.5 | 91.8 ± 3.5 | 92.4 ± 3.9 | 92.4 ± 4.0 |
| Learned Mean-Bound | $^\star$**95.0 ± 2.2** | $^\star$**94.9 ± 2.2** | 86.0 ± 3.4 | 85.9 ± 3.4 | 93.1 ± 3.4 | 93.1 ± 3.4 |
| Learned Max-Bound | 92.6 ± 5.6 | 92.6 ± 5.6 | **92.5 ± 1.7** | **92.5 ± 1.7** | **94.9 ± 1.1** | **94.8 ± 1.1** |
| *arXiv (3L-64U)* | | | | | | |
| Sentence Order | 87.8 ± 0.5 | 87.3 ± 0.5 | 85.5 ± 0.8 | 84.9 ± 0.9 | 85.9 ± 0.7 | 85.3 ± 0.7 |
| Window Co-occurrence | 87.9 ± 1.9 | 87.4 ± 2.0 | **85.8 ± 0.8** | **85.1 ± 0.9** | 87.0 ± 1.0 | 86.4 ± 1.1 |
| Max Semantic Similarity | 87.8 ± 0.8 | 87.3 ± 0.8 | 85.6 ± 1.0 | 84.9 ± 1.1 | 86.3 ± 0.9 | 85.7 ± 1.0 |
| Learned Mean-Bound (sample) | 88.2 ± 3.1 | 87.6 ± 3.2 | 84.1 ± 1.5 | 83.5 ± 1.5 | 88.4 ± 0.9 | 87.9 ± 1.1 |
| Learned Max-Bound (sample) | $^\star$**91.7 ± 1.9** | $^\star$**91.3 ± 2.1** | 83.5 ± 1.6 | 82.9 ± 1.6 | **89.9 ± 1.4** | **89.6 ± 1.3** |

## D    GRAPH CONSTRUCTION STRATEGY AND STORAGE

**Symmetrizing Attention Matrices.**    Although attention coefficients are learned in a directed manner, we transform the resulting attention matrix $\alpha = \text{Attn}(X) \in R^{L \times L}$ into an undirected weighted graph. After statistical filtering, we follow a row-wise operation. Given the row $i$, for each non-zero entry $\alpha_{ij}$, we introduce an edge $(i, j)$ with weight $w_{ij} = \alpha_{ij}$. To enforce symmetry, each edge induces its inverse edge by adding the edge $(j, i)$ with its corresponding weight. If subsequent rows in $\alpha$ reveal dependencies already calculated, we compare their attention coefficient and redefine the edge weight as

$$w_{ij} = w_{ji} = max(\alpha_{ij}, \alpha_{ji}). \tag{10}$$

This process ensures a consistent undirected representation while preserving the strongest dependencies between sentence nodes.

**Single-Pass Graph Construction.**    The resulting learned document graphs are precomputed and stored as PyTorch Geometric objects. Unlike alternative approaches constructing graphs on the fly, our implementation incurs the graph-creation cost only once, significantly reducing computational overhead by eliminating the need for graph reconstruction across epochs and model variations.

## E    GRAPH-BASED VS. NON-GRAPH APPROACHES

### E.1    CLASSIFICATION METHODS

While the focus of this work is on graph-based strategies for document representation and their impact on document classification tasks, we also provide a comparative overview of recent non-graph-based approaches utilizing traditional vector-based representations for document classification. Table 6 summarizes the performance of recently proposed models on the datasets considered in this paper.

Table 6: Classification results of our proposed learned graph structures compared to heuristic-based graph construction methods and recent non-graph approaches. Reported metrics include accuracy and macro-averaged $F_1$ score. Results marked with ‡ are not directly comparable, as they use a subsample of the arXiv dataset and only abstracts for classification.

| | BBC News | | HND | | arXiv | |
|---|---|---|---|---|---|---|
| Graph Scheme | Acc | $F_1$-ma | Acc | $F_1$-ma | Acc | $F_1$-ma |
| *Non-graph-based strategies* | | | | | | |
| Longformer-base | 97.9 | 97.8 | 85.4 | 85.3 | 86.9 | 86.7 |
| LongT5-tglobal-base | 96.3 | 96.3 | 74.6 | 74.5 | 87.8 | 87.8 |
| BERT (Park et al., 2022) | – | – | 92.0 | – | – | – |
| CogLTX (Park et al., 2022) | – | – | 94.8 | – | – | – |
| rRF (Singh et al., 2022) | 96.2 | 96.1 | – | – | – | – |
| ConfliBERT-SCR (Hu et al., 2022) | – | 98.1 | – | – | – | – |
| Prefix-Propagation (Li et al., 2023a) | – | – | – | 81.8 | – | 83.3 |
| LSG (Condevaux & Harispe, 2023) | – | – | – | – | – | 87.9 |
| RAN+Random (Li et al., 2023b) | – | – | 93.9 | – | 80.1 | – |
| RAN+GloVe (Li et al., 2023b) | – | – | 95.4 | – | 83.4 | – |
| RAN+Pretrain (Li et al., 2023b) | – | – | **96.9** | – | 85.9 | – |
| PFC (Yun et al., 2023) | 98.1 | 97.1 | – | – | ‡76.0 | ‡61.0 |
| RoBERTa (Reusens et al., 2024) | 98.0 | 97.0 | – | – | – | – |
| Llama-3.2-1B-Instruct (Li et al., 2025a) | – | – | – | – | 89.2 | 89.0 |
| Llama-3.2-3B-Instruct (Li et al., 2025a) | – | – | – | – | 90.4 | 90.3 |
| ModernBERT-base (Li et al., 2025a) | – | – | – | – | 81.0 | 81.1 |
| AChorDS-LVQ (Mohammadi & Ghosh, 2025) | – | – | 91.8 | – | – | – |
| *Heuristic-based graphs* | | | | | | |
| complete graph | **99.9** | **99.9** | 94.6 | 94.5 | – | – |
| sentence order | 99.7 | 99.7 | 92.6 | 92.6 | 87.8 | 87.3 |
| window co-occurrence | 99.8 | 99.8 | 92.1 | 92.1 | 87.9 | 87.4 |
| mean semantic similarity | 99.4 | 99.3 | 91.2 | 91.1 | – | – |
| max semantic similarity | 99.7 | 99.7 | 92.8 | 92.8 | 87.8 | 87.3 |
| *Our learned graphs* | | | | | | |
| learned mean-bound | **99.9** | **99.9** | 95.0 | **94.9** | – | – |
| learned max-bound | 99.6 | 99.6 | 92.6 | 92.6 | **91.9** | **91.7** |

Park et al. (2022) fine-tuned several Transformer-based models, including **BERT** (Devlin et al., 2018) and **CogLTX** (Ding et al., 2020). BERT was fine-tuned on truncated inputs to the first 512 tokens, using a fully connected layer on the [CLS] token for classification. In turn, the Cognize Long TeXts (CogLTX) model was included in the study with the hypothesis that a small set of key sentences is sufficient for accurate document classification.

Another method, **rRF** (removal of Redundant Feature) (Singh et al., 2022) applies dimensionality reduction by eliminating redundant information based on word-level similarity scores computed using GloVe embeddings (Pennington et al., 2014), followed by a Naive Bayes classifier.

**ConfliBERT** (Hu et al., 2022) is a domain-specific pre-trained language model for conflict and political violence detection. Although the authors explore both pretraining from scratch and continual pretraining strategies, Table 6 only reports the best-performing variant – pretrained from scratch using cased data (SCR).

Although parameter-efficient tuning methods aim to reduce memory overhead while attaining comparable performance to fine-tuning of pretrained language models, they often fail to model long documents. To address this, Li et al. (2023a) propose **Prefix-Propagation**, a technique that allows prefix hidden states to dynamically evolve across layers by incorporating them into the attention mechanism.

To further mitigate the quadratic complexity of Transformer self-attention for long sequences, Local Sparse Global (**LSG**) attention is proposed in (Condevaux & Harispe, 2023). LSG follows a block-based processing of the input and applies local attention to capture local context for nearby dependencies, sparse attention for extended context, and global attention to improve information flow inside the model.

In a similar direction, Li et al. (2023b) propose the Recurrent Attention Network (**RAN**), which introduces a recurrent formulation of self-attention to handle long sequences, enabling long-

term memory and extracting global semantics in both token-level and document-level representations. RAN processes sequences in non-overlapping windows, applying positional multi-head self-attention to a window area, and propagates a global perception cell vector across windows to capture long-term dependencies. Table 6 presents results for three RAN variants: i) RAN+Random, with randomly initialized weights; ii) RAN+GloVe, using GloVe embedding (Pennington et al., 2014) as word representation; and iii) RAN+Pretrain, pretrained with a masked language modeling objective on the BookCorpus (Zhu et al., 2015) and C4 (RealNews-like subset) (Raffel et al., 2020).

To further reduce the computation of self-attention, Yun et al. (2023) propose a **PFC** strategy, which integrates a token pruning step to eliminate less important tokens from attention computations, and a token combining step to condense input sequences into smaller sizes.

Despite such innovations, full model fine-tuning remains widely adopted in document classification. For instance, a fine-tuned **RoBERTa** (Liu et al., 2019) was used by Reusens et al. (2024), combining Bayesian search with author recommendations for hyperparameter setting. Similarly, Li et al. (2025a) evaluate small language models in real-world classification tasks, focusing on best practices and tuning strategies to address text classification effectively. The study included **Llama3.2 (1B-3B)** (Touvron et al., 2023) and **ModernBERT-base** (Warner et al., 2024).

Finally, Adaptive Chordal Distance and Subspace-based LVQ (**AChorDS-LVQ**) (Mohammadi & Ghosh, 2025) is introduced as a prototype-based approach for learning on the manifold of linear subspaces derived from input vectors. The method learns a set of subspace prototypes to represent class characteristics and relevance factors, automating the selection of subspace dimensionalities and the influence of each input vector on classification outcomes.

### E.2 CLASSIFICATION RESULTS

In both the BBC News and arXiv datasets, our learned graph structures consistently outperform all baseline models, including both heuristic-based graphs and recent non-graph approaches. On BBC News, our learned mean-bound graphs achieve near-perfect performance with 99.9% accuracy and $F_1$ score, significantly surpassing the best non-graph alternative, PFC, which reaches 98.1% accuracy and 97.1% $F_1$ score. Similarly, on arXiv, our learned max-bound graphs have a considerable advantage over other graphs as well as over the strongest non-graph model, fine-tuned Llama-3.2. While Llama-3.2 reports 90.4% accuracy for the 3B version and 89.2% accuracy for the 1B variant, our learned graphs yield 91.9% accuracy and 91.7% $F_1$ score without requiring manual constructions or task-specific expert knowledge. In contrast, on the HND dataset, heuristic-based graph methods underperform compared to non-graph baselines. However, our learned graphs remain competitive with top-performing models, such as RAN and CogLTX, demonstrating their capacity to capture the document structure.

The observed results underscore the effectiveness of automatically identifying task-relevant segments within input sequences, supporting the integration of local contextual information at lower textual granularities while preserving global semantics at higher levels. Moreover, the performance of RAN demonstrates the benefit of attention mechanisms that operate over windows with explicit propagation of information from fine-grained units (e.g., tokens) to higher-level representations. Such a strategy offers a clear advantage over conventional sequential models in constructing comprehensive document representations. The results from Table 6 further motivate future work to explore alternative filtering strategies, other attention mechanisms, and hierarchical approaches to constructing graphs over multiple text granularities (e.g., sentences, sections) via heterogeneous graph structures.

**Use of Large Language Models.** AI assistants were only used for paraphrasing and spell-checking. All content, ideas, and claims presented in this paper remain the original work of the authors.

