# OpenReview forum: "Rethinking Graph-Based Document Classification: Learning Data-Driven Structures Beyond Heuristic Approaches"
_ICLR.cc/2026/Conference — ICLR 2026 Conference Withdrawn Submission_

### Official Review · Reviewer_4Lio · 2025-10-31

**Soundness:** 1
**Presentation:** 2
**Contribution:** 1
**Rating:** 2
**Confidence:** 4

**Summary:**

This paper aims to introduce a graph-attention framework for automatically constructing a homogenous graph by leveraging self-attention between sentence representations with bounding methods. Overall, the porposed method lack of novelty and in-depth analysis.

**Strengths:**

1. This paper pointed out some limitations of existing works on text classification, especially graph-based frameworks.
2. The proposed framework could achieve better performance on document classification tasks on selected benchmark dataset.

**Weaknesses:**

- Lack of Novelty:
The proposed framework applies self-attention to model correlations between sentences within a document, which is a well-established approach. Moreover, the handling of repeated sentences ignores contextual information, and the method treats sentence order in a bag-of-words manner without modeling reading sequence.

- Limited Evaluation:
The experimental validation is insufficient. The framework is only tested on limited text classification settings. Broader evaluation is expected across diverse tasks, including mono-label/multi-label classification, sequence-level and word-level classification, and natural language inference (NLI).

- Insufficient Analysis:
The paper lacks in-depth analysis demonstrating how sentence correlations are captured. Additional analysis is needed to validate whether the proposed method effectively models inter-sentence relationships.

**Questions:**

Have you tested the framework on more diverse tasks (e.g., multi-label or NLI) to demonstrate generality?
Can you provide visualizations or attention maps showing how sentence correlations are captured?
Why was the decision made to process identical sentences in isolation without leveraging surrounding context?
Could modeling positional or sequential sentence information improve performance or interpretability?

---

> ### Author Response · Authors · 2025-12-01
> **Addressing Main Concerns: Novelty, Order Modeling, and Evaluation Scope**
>
> We thank the reviewer for noting the strengths of our work, including its performance improvements and the clear articulation of limitations in prior graph-based document classification approaches. We address the concerns and questions below.
>
> ---
>
> ### __(1) Novelty Concerns__
> We respectfully emphasize that our contribution extends far beyond applying self-attention to sentence embeddings. While self-attention as a feature extractor is well established, our work introduces **a new paradigm for document-level graph structure learning.**
>
> __Limitations of traditional document graphs:__
> Prior approaches rely on heuristic rules–fixed windows, discourse markers, shallow semantic cues, entity graphs, or paragraph boundaries–which: (i) require substantial domain expertise, (ii) produce hand-designed task-agnostic topologies, and (iii) do not adapt to document semantics.
>
> In turn, __our contributions introduce a different formulation:__
> * We **learn the graph structure directly from data,** rather than fully relying on heuristics.
> * Unlike prior GSL methods, our model employs a **dedicated graph-learning attention module independent of the downstream GNN classifier**, allowing the learned structure to generalize across architectures.
> * We propose **a variance-aware statistical filtering mechanism** that transforms dense attention matrices into sparse, semantically meaningful document graphs.
> * The resulting graphs consistently benefit multiple GNN backbones (`GAT, GCN, GraphSAGE`), demonstrating that the improvements derive from the graph-construction module itself, not from any particular encoder.
>
> To the best of our knowledge, **decoupling graph induction from graph classification for document-level tasks** in this manner has not been explored in prior work. Our expanded ablations (included in the revised Appendix C) further validate that the model’s gains come from this novel pipeline rather than from standard attention or GAT components.
>
> ---
>
> ### __(2) Order modeling__
> We clarify that **our method does not discard sequential information.**
> * Each sentence embedding is produced by a Sentence Transformer, which inherently uses positional embeddings; thus, sequential context informs each sentence-level representation.
> * The statistical filtering then determines whether adjacent sentences carry task-relevant relationships. For long-document classification, cross-section and thematic connections often outweigh strictly local adjacency, in line with prior findings in hierarchical text modeling.
>
> To assess the value of explicit sequential structure, **we included a sequence-order graph baseline**, which systematically underperformed our learned graphs across all datasets. This supports the conclusion that the semantic relations captured by our learned edges provide stronger supervision than fixed adjacency patterns.
>
> ---
>
> ### __(3) Evaluation Scope__
> We appreciate the encouragement to expand the empirical study. Due to the limited rebuttal period and the computational cost of (i) training the self-attention model used for graph induction, (ii) constructing multiple heuristic graph variants, and (iii) training several GNN and non-graph baselines, adding entirely new datasets during rebuttal is not feasible.
>
> However, our current evaluation already covers **two domains (news and scientific articles)**, spans **medium to extremely long documents**, and includes **different label granularities**. The **consistent gains across these diverse settings** indicate that our method is not dataset-specific. Moreover, improvements become stronger for long documents, suggesting that long-form datasets are especially meaningful.
>
> *As noted in our responses to Reviewer HA2B, planned extensions include `PubMed`, `MIMIC-III`, and other `extractive summarization datasets`.* We believe these additions will meaningfully broaden domain and task coverage in future work.

---

> > ### Author Response · Authors · 2025-12-01
> > **Clarifications on Relation Analysis, Duplicate Sentence Handling, and Sequential Modeling**
> >
> > ### __(4) Analysis of Inter-Sentence Relations__
> > While Figure 4 illustrated filtered adjacency matrices for representative samples, **we have now included raw head-wise and averaged attention heatmaps** for randomly selected samples from each dataset (Appendix B.3). These visualizations clearly reveal non-trivial, semantically meaningful cross-sentence dependencies, and they make explicit the need for filtering diffuse or noisy attention patterns as our statistical filtering component does.
> >
> > ---
> >
> > ### __(5) Handling of Identical Sentences__
> > We confirm that while duplicate sentences are unified into a single node for efficiency, **their contextual information is not lost**. Each occurrence contributes its own self-attention-derived edges, which are aggregated into the final edge set for the shared node. Thus, if a sentence appears twice in different contexts, both sets of contextual relations are preserved in the induced graph.
> > This strategy reflects a design choice grounded in document-level tasks, where **global semantic structure typically outweighs fine-grained positional distinctions.**
> >
> > ---
> >
> > ### __(6) Potential Gains from Explicit Sequential Modeling__
> > We agree that, in principle, explicit sequential modeling could improve performance. To test this, **we included sentence-order graphs among our baselines**. Across all datasets, these order-based graphs were consistently outperformed by our learned graphs, indicating that **semantic dependencies dominate the useful signal for document-level classification**. This finding is compatible with prior work in long-document modeling, where global thematic structure tends to be more discriminative than local adjacency.
> >
> > ---
> >
> > We hope these clarifications illustrate the conceptual novelty, methodological soundness, and empirical rigor of our approach. We believe the presented evidence meaningfully strengthens the case for our contributions and addresses the reviewer’s concerns.

---

### Official Review · Reviewer_HA2B · 2025-10-31

**Soundness:** 3
**Presentation:** 3
**Contribution:** 1
**Rating:** 2
**Confidence:** 4

**Summary:**

In the submitted manuscript, the authors propose to sparsify the attention matrix (summed over the heads) of a pretrained self-attention model for document classification. The sparsified attention matrices are then fed to a Graph Neural Network to improve document classification performance. The proposed method is evaluated on three benchmark datasets and compared to several heuristic graph constructions and a variety of transformer baselines.

**Strengths:**

- The proposed graph inference method is indeed more adaptable to different datasets than the different heuristic graph constructions that are listed by the authors.

- The overview Figure 2 provides a good understanding of the approach.

**Weaknesses:**

- Evaluating on only three datasets is not a lot and the transformer baselines you use change for each dataset. I think the empirical evidence in favour of your method should be extended for the method to really be of proven practical relevance.

- I am not convinced that the methodological contribution or the empirical work offer sufficient novelty to warrant publication at the ICLR conference.

- You say that heterogeneous graphs are "not comparable" to your homogenous construction and are therefore not included as baselines. I am not convinced by this. I do not see why the performance of these methods is not compared. In fact, it seems to me that too few of the recent graph-based approaches to document classification are included as baselines.

- The results in Table 2 seem to indicate that the proposed graph construction is rather memory intensive.

**Questions:**

1] It is unclear to me how the multi-head self attention layers, that give rise to your graph construction, are trained. Are these also trained on the document classification task? And do you sum over the attention matrices of all layers or do you use the attention matrices of a particular layer of your self-attention model?

2] What is the intuition of using the standard deviation term in your sparsifying mechanisms in Equations (1) and (2), i.e., why would you like to distinguish rows with high or low variance attention scores in the attention matrices in your sparsifying mechanism?

3] Concerning your experiments, I have the following questions:

3.1] It is unclear to me why the Transformer baselines change for each dataset. Could you please include the performance of each of these transformers on the datasets? I furthermore want to encourage you to include the performance of state-of-the-art transformers, i.e., the best known performance of language models on these datasets.

3.2] I think seemingly arbitrarily fixing the window size to 3 for the window-based co-occurrence graph constructions may not result in a fair comparison. Did you also evaluate the performance of models based on these graph constructions for larger window sizes?

3.3.1] I am not sure whether the models in your experiments have converged. Training the self-attention model for 20 epochs and the GAT model for 50 epochs seems insufficient to me. Could you include the loss curves of some of your experiments in the appendix to evidence that this amount of training is sufficient?

3.3.2] I am in particular wondering whether your results in Table 3 still look similar if your self-attention mechanism is trained for more epochs? If the sparse attention pattern that you produce with your filtering is an optimal message passing template, should the attention mechanism not learn it if trained to convergence?

3.4] For conclusive results it would be better if the depth and embedding dimensions of your GAT model were the result of a grid search instead of fixed as in Line 403.

4] Minor Comments:

4.1] In Line 164-5 you state the Graph Structure Learning methods "often yield incomplete or task-misaligned topologies". Could you provide a reference or empirical results to substantiate this claim?

4.2] In Line 202 you define the edge set to be a set of scalars. This is rather uncommon. Usually edge sets are defined to be sets of node tuples.

4.3] In Line 248 you mention that "self-loops are explicitly removed" in your graph construction. It is rather uncommon to fit GNNs without self-loops. Is it true that you add self-loops back into the graphs in order to run your GNN classifiers on them?

4.4] In Line 392 you mention that you use average pooling in your GAT model. Often in graph-level classification tasks, one observes sum pooling to yield better-performing models. I am not sure whether you experimented with this pooling operator as well?

4.5] Did you ablate the impact of aggregating your attention matrices across heads? In other words, does the performance of your method change if only individual heads are used in the GNN?

---

> ### Author Response · Authors · 2025-11-24
> **Clarifications on Weaknesses  --- Questions will be addressed in a separate comment**
>
> We thank the reviewer for the detailed and constructive feedback and for highlighting the adaptability of our graph inference method and the clarity of Fig. 2. We address all concerns below.
>
> ---
>
> ### __New datasets__
> We agree that expanding the empirical evaluation is valuable. However, due to the limited rebuttal period and the computational cost of (i) training the self-attention model used for graph induction, (ii) constructing multiple heuristic-based graphs, and (iii) training several downstream GNNs and non-graph baselines for consistent comparison, adding new datasets is not feasible during rebuttal.
>
> Our current evaluation already spans different domains--news and scientific articles--with document lengths ranging from medium to very long, and with varying label granularities. The consistent gains across these settings indicate that our method is not dataset-specific. Notably, improvements become stronger for long documents, suggesting that additional long-form datasets are more meaningful than standard short-text benchmarks.
>
> Planned extensions include `PubMed` (~3.2k-word scientific papers), `MIMIC-III` (clinical multi-label classification, subject to data-use agreements), and `extractive summarization datasets` with sentence-level annotations. These additions will expand the domain and task coverage in future work.
>
> ---
>
> ### __Novelty concerns__
> We appreciate the opportunity to clarify our contributions:
> + **A decoupled graph learning module** that constructs sentence-level edges independently of the downstream GNN encoder, unlike prior GSL work that jointly learns graph structure and graph classifier.
> + **A statistical variance-aware sparsification mechanism** (mean/max filtering) that transforms noisy attention scores into stable document graphs.
> + **Generalizability across backbones**, with consistent gains on `GAT, GCN`, and `GraphSAGE`, showing that our approach is not tied to any particular graph encoder (Appendix C; _also discussed with Reviewer Eg8B_).
>
> Our setting fundamentally differs from existing GSL methods for text, which learn corpus-level graphs, ignoring intra-document structure and fine-grained semantics. We instead introduce a **stand-alone graph learner** inducing sentence-level relational graphs grounded in document semantics. This constitutes the core novelty of our method.
>
> ---
>
> ### __Excluding heterogeneous graph baselines__
> Our intent was not to dismiss heterogeneous models but to explain why they do not offer a fair or comparable baseline in our setting:
> + **Heavy reliance on heuristics and limited reproducibility:** heterogeneous methods depend on predefined node types (topics, entities, vocabulary units) and curated edge schemas. Implementations of representative models (e.g., Text-MGNN[1], FT-GCN[2], RoBERTaGCN[3]) are not publicly available and require reconstructing undocumented preprocessing pipelines or manually curated resources not included in standard datasets. This introduces significant variance and undermines fairness.
> + **Framework incompatibilities:** Many heterogeneous models [4,5] are implemented in TensorFlow or DGL. Porting their components to PyTorch Geometric--the framework we used--would require reimplementation, making differences attributable to engineering rather than modeling choices.
>
> To avoid conflating model quality with graph-type differences, we focus on strong homogeneous baselines. We will clarify this in the revised paper.
>
> ---
>
> ### __Memory cost concerns__
> We agree that memory efficiency is important. Dynamic graph construction has been explored in the literature, but it typically incurs significant computational overhead as graph size increases and may introduce concurrency issues during updates. Although constructing graphs on the fly reduces disk usage, **pre-storing graphs avoids repeated computation during GNN training.**
> In our case, i) graph construction is performed only once before training, and ii) the disk usage of our learned graphs is comparable to simple heuristic graphs, suggesting that memory costs are dominated by **node embeddings**, not edges, which store only single-value attributes.
>
> Future work may explore reducing node counts through graph compression [6] or strategically reducing node embedding dimensionality. We will include this discussion in the revised version.

---

> > ### Author Response · Authors · 2025-11-24
> > **References (Clarifications on Weaknesses)**
> >
> > >1. Gu, Y., Wang, Y., Zhang, H. R., Wu, J., & Gu, X. Enhancing text classification by graph neural networks with multi-granular topic-aware graph. IEEE Access, 2023.
> > 2. Cai, H., Lv, S., Lu, G., & Li, T. Graph convolutional networks for fast text classification. ICNLP, 2022.
> > 3. Nakajima, H., & Sasaki, M. Text classification using a graph based on relationships between documents. PACLIC, 2022.
> > 4. Ragesh, R., Sellamanickam, S., Iyer, A., Bairi, R., & Lingam, V. Hetegcn: heterogeneous graph convolutional networks for text classification. WSDM, 2021.
> > 5. Lin, Y., Meng, Y., Sun, X., Han, Q., Kuang, K., Li, J., & Wu, F. BertGCN: Transductive Text Classification by Combining GNN and BERT. Findings of ACL-IJCNLP, 2021.
> > 6. Jang, Y., Kim, D., & Ahn, S. Graph generation with $ k^ 2$-trees. arXiv, 2023.

---

> > > ### Author Response · Authors · 2025-11-30
> > > **Answers to Reviewer Questions**
> > >
> > > **1. Training of self-attention model:** Yes. The model used for graph induction is trained on the document classification task, allowing it to learn task-specific sentence dependencies. For graph construction, we average the attention matrices across heads to incorporate complementary relational patterns learned by each head. We have included in Appendix B.3 visual comparisons between head-averaged and head-specific attention matrices for randomly selected samples from each dataset. Such figures confirm that head-averaging produces more stable and representative structures.
> > >
> > > **2. Intuition behind using standard deviation in sparsification:** The standard deviation captures how decisively a sentence attends to others. Therefore, a high variance means that the sentence focuses strongly on a few informative targets, while a low variance suggests that attention is diffuse and semantically weak.
> > > Incorporating the standard deviation therefore allows our filter to normalize such differences and select edges corresponding to meaningful dependencies. As shown in newly added Figures 6–8 (Appendix B.3), some rows in the averaged-attention matrices exhibit significantly higher variance than others, validating this intuition: low-variance rows correspond to flat, largely non-decisive patterns, whereas high-variance rows highlight strong semantic correlations.
> > >
> > > **3.1. Transformer baselines and consistency across datasets:** In the original submission, we reported results directly from prior work on each dataset; hence the baselines differed. To remove this inconsistency, we now provide two Transformer baselines fine-tuned for each of the datasets: `Longformer-base-4096`, with a 4k-token window, and `LongT5-global-base`, supporting up to 16k tokens.
> > > Both models were fine-tuned under consistent settings. Our **learned graphs outperform these strong baselines across datasets**. Additional comparisons with prior non-graph models and further LMs are included in Appendix E.
> > >
> > > **3.2. Window size for co-occurrence graphs:**
> > > We chose window size 3 to ensure graph densities comparable to our learned graphs. Among heuristic baselines, each method spans a different edge-count range; using w=3 places window graphs in the middle of this spectrum and avoids extreme sparsity or over-connectivity.
> > > Notably, using a window-based graph with w=2 would result in sentence-order chains, while a larger window (w=4) produce denser graphs--averaging roughly 104 (BBC), 105 (HND), and 3050 (arXiv) edges--making them incomparable to our learned graphs and biasing the comparison toward denser structures.
> > > Moreover, **window size 3 aligns with recommendations from prior work** [7,8], which report that mid-range windows achieve better performance for document graphs.
> > >
> > > **3.3. Model convergence and effect of training longer:**
> > >
> > > * __Regarding model convergence:__ We have included Figure 5 (Appendix B.2), which provides the learning curves of the self-attention model later used for graph induction on arXiv. We train with early stopping based on validation macro-F1 (patience = 5), with a maximum of 20 epochs. In practice, training consistently converged between epochs 8 and 11, and increasing the maximum epoch count (as well as patience) did not improve final classification performance in preliminary experiments.
> > >
> > > * __Regarding sparsity:__ We argue that even under extended training, the raw attention matrices remain inherently dense and noisy. While additional optimization may slightly sharpen the distribution, attention weights typically retain widespread non-zero activations, preventing the emergence of a meaningful sparse topology. Our ablations (Table 3, Section 5) show that applying our statistical filtering mechanism yields markedly stronger classification performance and induces substantially more efficient graph structures. In contrast, graphs derived directly from unfiltered attention (raw learned weights) perform significantly worse. This establishes the filter as a critical component for producing high-quality edge sparsification, lightweight graph representations (fewer edges stored on disk), and consistent improvements in downstream accuracy. Although attention serves as the backbone of our graph-induction framework, **the statistical filter is indispensable for obtaining robust document-level graph representations** for classification.
> > >
> > > **3.4. Grid search for GAT depth and dimensions:** As detailed in Section 4.3 (_“Graph Attention Network (GAT)”_), we perform a grid search over GNN depth `{1, 2, 3}` and hidden dimensions `{64, 128, 256}`. Section 5 (and Table 2) reports only the best configuration per dataset. Thus, **the hyperparameters used in our main results are selected from this grid search**.
> > >
> > > > 7. Huang et al. Text Level Graph Neural Network for Text Classification. EMNLP-IJCNLP, 2019.
> > > > 8. Zhang et al. Every Document Owns Its Structure: Inductive Text Classification via Graph Neural Networks. ACL, 2020.

---

> > > > ### Author Response · Authors · 2025-11-30
> > > > **Addressing Minor Comments**
> > > >
> > > > **4.1. “Incomplete or task-misaligned topologies” of GSL methods:** Thanks for pointing this out. The statement refers specifically to heuristic-based graph constructions, not GSL methods. We have revised the text for clarity.
> > > >
> > > > **4.2. Edge set notation:** We agree and have revised the notation to define edges as node tuples with associated weights. Thanks for your suggestion!
> > > >
> > > > **4.3. Handling of self-loops:** Correct. We remove self-loops during graph construction but reintroduce them automatically within the GNN encoder by adding the identity matrix to the adjacency structure (with weight 1.0). We avoid retaining diagonal attention weights in our generated graphs because they were consistently low; incorporating them directly would inject weak self-information relative to neighboring messages.
> > > >
> > > > **4.4. Sum pooling vs. mean pooling:** We tested sum pooling in preliminary experiments on BBC News and HND. Across both datasets, **sum pooling reduced classification performance** (by approximately 0.1--0.3 F1 for max-filter graphs and up to 2.3 F1 for the mean-filter variant). **Conversely, mean pooling provided more stable and competitive results**, which is why we adopted it in the main experiments.
> > > >
> > > > **4.5. Ablation of attention aggregation across heads:** We performed a preliminary qualitative inspection of head-wise attention patterns and observed substantial variation across heads and datasets. Aggregating across heads produced more stable and semantically coherent relational structures. Figures 6--8 (Appendix B.3) illustrate these differences. While we did not perform a head-by-head quantitative ablation, **the qualitative evidence strongly supports aggregation as the most robust choice**.

---

### Official Review · Reviewer_QLiv · 2025-11-01

**Soundness:** 2
**Presentation:** 2
**Contribution:** 2
**Rating:** 2
**Confidence:** 4

**Summary:**

This paper presents a data-driven graph structure learning approach for document classification, aiming to replace traditional heuristic graph constructions (e.g., window-based co-occurrence, sequential order, semantic similarity). Experiments on BBC News, Hyperpartisan News Detection (HND), and arXiv datasets show improvements of up to 4.3 F1 points over heuristic graphs and moderate gains over small LLM baselines.

**Strengths:**

- Simplicity and generality.
- Empirical evidence.

**Weaknesses:**

- Limited task scope.
- Weak theoretical justification.

**Questions:**

1. How does this approach differ from a joint GAT + attention-based GSL baseline?
2. The mean- and max-bound thresholds (Eq. 1–2) reintroduce heuristics, contradicting the claim of “no heuristic design.” Threshold $\delta$ is a manually tuned hyperparameter (Table 3), and its behavior differs across datasets.
3. No direct comparison to models using the same Sentence Transformer embeddings without graph construction.
4. While inspired by attention mechanisms, the authors provide no theoretical grounding for why attention-derived edges approximate semantic or functional dependencies.
5. Traditional GSL senarios often compensate for weak semantic features by learning graph structures. However, this paper aims to build semantically rich document graphs. In this context, is explicit GSL still necessary? A pretrained LLM may already capture the same relational semantics more effectively.

---

> ### Author Response · Authors · 2025-11-22
> **Clarifications on Model Design, Filtering Strategy, and Baseline Comparisons**
>
> We thank the reviewer for the constructive feedback and for recognizing the simplicity and generality of our method, as well as its empirical strengths. We address each point concisely below.
>
> ---
> ### __(1) Difference from a joint GAT + attention-based GSL__
>
> Our approach is **not** a joint GNN-GSL model. Unlike joint approaches, where attention is optimized together with the GNN and thus entangles graph learning with GNN classifier bias, our graph learner is __fully decoupled__:
>
> * It computes pairwise sentence dependencies using an independent self-attention layer.
> * The graph is **computed once and fixed** before any GNN training.
> * Any GNN (e.g., GCN, GAT, GraphSAGE) can be plugged in **without retraining** the graph generator — _New experiments with multiple GNNs (Appendix C) confirm that all benefit similarly. Results were also provided in a separate official comment to reviewer Eg8B._
>
> This decoupling 1) avoids feedback loops where the GNN-classifier biases the learned graph and 2) ensures architectural generality. We have clarified this distinction more explicitly in Section 2.2.
>
> ---
> ### __(2) Concern about _heuristics_ in the statistical filtering__
>
> We clarify that the filtering procedure is not heuristic in the sense used in GSL literature. It does not incorporate domain rules, fixed windows, syntactic templates, or handcrafted assumptions. Instead:
>
> * The mean/max filters follow standard score normalization used in attention literature (e.g., in pruning and adaptive sparsification).
> * The sparsity level is determined by $\delta$, which is a single scalar hyperparameter functioning similarly to dropout rate or weight decay.
> * Our paper includes a sensitivity analysis on HND showing that **performance is stable across reasonable $\delta$ ranges**.
>
> Thus, our filtering is **statistical and data-driven**, rather than heuristic in the traditional sense.
>
> ---
> ### __(3) Comparison to Sentence Transformer alone__
> We thank the reviewer for this suggestion. We have now added: `SentenceEmb + MLP`, a direct baseline representing each document as the mean of its Sentence Transformer sentence embeddings, followed by a two-layer MLP classifier, and `ReLUAttn-SentenceEmb + MLP`, which corresponds to our self-attention models later used for graph generation (Section 3.2). Results (provided in Table 2 of our revised paper) show that:
> * While these models perform well on short documents (BBC News), they degrade sharply on HND and arXiv.
> * Although ReLUAttn shows improvements over SentenceEmb + MLP, they remain significantly below graph approaches.
> * Our learned graphs consistently outperform pure embedding baselines, even when trained with only 10% of the data, as on arXiv.
>
> These findings support the conclusion that structural information from inter-sentence relations is crucial and cannot be captured by pooled embeddings alone.
>
> ---
> ### __(4) Attention-derived edges without theoretical guarantees__
> Our approach is empirically motivated, following established findings that show self-attention heads frequently align with interpretable linguistic patterns, such as syntactic dependencies or positional functions [1,2], and that inter-sentence attention can successfully model discourse relations [3].
> Our contribution is not a new semantic theory, but:
>
> * A general and empirically validated graph-generation mechanism,
> * A statistical filtering method that stabilizes attention-based relations into meaningful and sparse graph structures,
> * Extensive evidence that these learned graphs lead to consistent and significant gains across datasets and GNN backbones (Appendix C).
>
> > [1] Voita, E., Talbot, D., Moiseev, F., Sennrich, R., & Titov, I. Analyzing Multi-Head Self-Attention: Specialized Heads Do the Heavy Lifting, the Rest Can Be Pruned. ACL 2019.
>
> > [2] Clark, K., Khandelwal, U., Levy, O., & Manning, C. D. What Does BERT Look at? An Analysis of BERT’s Attention. 2019 ACL, Workshop BlackboxNLP.
>
> > [3] Song, W., Song, Z., Fu, R., Liu, L., Cheng, M., & Liu, T. Discourse self-attention for discourse element identification in argumentative student essays. EMNLP 2020.
>
> ---
> ### __(5) Why use GSL if semantic encoders are strong?__
> Sentence encoders provide strong __independent__ sentence representations but lack explicit modeling of document-level relations. In contrast, our learned graphs help identify relevant relationships between sentences, propagate information across the document in a structured manner, and capture long-range dependencies—characteristics particularly important for long documents. This explains why our method outperforms Sentence Transformer baselines and small LLMs.
>
> ---
>
> We thank the reviewer again for the helpful suggestions. We have incorporated the requested baselines, clarified modeling distinctions, and strengthened empirical evidence.

---

### Official Review · Reviewer_Eg8B · 2025-11-01

**Soundness:** 2
**Presentation:** 2
**Contribution:** 1
**Rating:** 2
**Confidence:** 3

**Summary:**

This paper constructs homogeneous weighted graphs with sentences as nodes, while edges are learned via a self-attention model
that identifies dependencies between sentence pairs.

**Strengths:**

1. This paper introduces a self-attention-based approach that eliminates the dependency on heuristics and domain expertise.

2. Experiment results have verified the   the effectiveness of proposed model.

**Weaknesses:**

1. The novelty is not enough as only applying graph attention neural networks to document graph tasks.

**Questions:**

1. what is the differnce among your mehtod compared with graph attention neural network.

---

> ### Author Response · Authors · 2025-11-20
> **Clarifications on novelty, difference from standard GAT, and additional experiments.**
>
> We thank the reviewer for the constructive feedback and for recognizing the strengths of our work, including the removal of heuristic dependencies and the demonstrated empirical effectiveness. Below, we clarify the core novelty of our approach and address the questions that have been raised.
>
> ---
>
> ### **(1) Novelty Beyond Applying GAT to Document Graphs**
>
> We would like to emphasize that **our contribution is not the use of GAT for document classification**, but rather **the introduction of a learnable, data-driven graph construction mechanism.** The key idea of our work lies in **learning the document graph itself**, independently of the graph neural network used downstream.
>
> Most existing approaches--whether using GCN, GAT, or other GNN variants--rely on manually constructed edges (TF-IDF similarity, sliding windows, topic or entity heuristics, etc.).
> In contrast, **our approach fundamentally shifts the paradigm** by proposing:
> 1. A self-attention–based graph construction module that learns inter-sentence dependencies directly from data, independently of the final classifier (graph encoder).
> 2. A statistical filtering mechanism that selects statistically reliable sentence–sentence relationships and suppresses noisy or spurious edges.
> 3. Homogeneous weighted graphs that remove expert rules and domain knowledge, simplifying their application across datasets.
>
> Together, these components form **a trainable**, **data-driven alternative to heuristic or rule-based edge creation**, which constitutes the primary novelty of our work.
> To make this explicit in the revised manuscript, we will add a clearer discussion contrasting our approach with prior GAT-based document models.
>
> ---
>
> ### **(2) Clarification: Difference From Standard GAT**
>
> A standard GAT **assumes the graph is given** and performs attention-based aggregation only over predefined neighbors. Our method fundamentally differs because **we learn the graph itself**, replacing the adjacency matrix entirely through:
>
> * Self-attention over all sentence pairs → dense dependency scores
> * Statistical significance over attention distributions → sparse, filtered graph
> * Learned edge weights that reflect semantic relationships
>
> This leads to the following distinction:
>
> | Component | Traditional GAT Pipeline | Our Model |
> |-----------------|----------------------------------|----------------|
> |Graph construction| Heuristics or rules | Learned from data via self-attention |
> |Edge sparsification | Manual thresholds | Statistical filtering |
> |Edge weights| Often binary or heuristic | Learned weights |
> |Domain dependence | High| Low (no rules)|
>
> Thus, our approach is **orthogonal to GAT**. The GAT layer is merely one possible encoder. The learned graph can be paired with any GNN architecture.
>
> To reinforce this point, we expanded our experiments to include **GCN** and **GraphSAGE**, whose results are described below.
>
> ---
>
> ### **(3) Additional Experiments (GCN and GraphSAGE Variants)**
> The reviewer correctly noted that our initial experiments used GAT. We have now conducted an extensive set of additional experiments using:
> * Learned Graphs + GCN
> * Learned Graphs + GraphSAGE
> * Heuristic Graphs + GCN
> * Heuristic Graphs + GraphSAGE
>
> The results show that:
> * **Heuristic graph constructions are highly unstable across GNN backbones** (a heuristic that performs well with GAT may perform poorly with GCN or vice-versa).
> * Window-based co-occurrence in particular exhibits strong variance across datasets and models.
> * **In contrast, our learned graph consistently performs strongly across all three GNN architectures (GAT, GCN, GraphSAGE)**
>
> These findings demonstrate that: **1) our method is not specialized for GAT**, **2) graph learning is the source of improvement**, not attention inside the GNN encoder, and **3) the approach is modular and general**, and can be attached to any GNN.
>
> We have added these results to the paper (Appendix C), along with a discussion that highlights graph encoder-agnostic robustness.
> We also share the results of our new experiments below, in a separate comment.
>
> We thank the reviewer again for the comments, and we hope the provided clarifications and new results demonstrate the value and novelty of our contribution.

---

> > ### Author Response · Authors · 2025-11-20
> > **Additional Experiments**
> >
> > Note that the best results are indicated in __bold__, the second-best results are denoted with an asterisk (*), and the optimal setting for each dataset is enclosed in [brackets].
> >
> > > #### 1. Results on __BBC News__ (2L–64U)
> >
> > | Scheme | GAT Acc | GAT F1-ma | GCN Acc | GCN F1-ma | GraphSAGE Acc | GraphSAGE F1-ma |
> > |--------|---------:|-----------:|----------:|------------:|----------------:|------------------:|
> > | Complete Graph | __[99.9 ± 0.1]__ | __[99.9 ± 0.1]__ | __99.8 ± 0.1__ | __99.8 ± 0.1__ | *99.5 ± 0.3 | *99.4 ± 0.3 |
> > | Sentence Order | 99.7 ± 0.3 | 99.7 ± 0.4 | 99.1 ± 0.4 | 99.1 ± 0.4 | __99.6 ± 0.2__ | __99.6 ± 0.2__ |
> > | Window Co-occurrence | *99.8 ± 0.3 | *99.8 ± 0.4 | 98.4 ± 1.1 | 98.3 ± 1.2 | 99.0 ± 0.5 | 98.9 ± 0.6 |
> > | Mean Semantic Similarity | 99.4 ± 0.5 | 99.3 ± 0.6 | 99.4 ± 0.5 | *99.4 ± 0.5 | 98.8 ± 0.6 | 98.8 ± 0.7 |
> > | Max Semantic Similarity | 99.7 ± 0.2 | 99.7 ± 0.2 | 99.1 ± 0.8 | 99.1 ± 0.9 | 99.4 ± 0.2 | *99.4 ± 0.2 |
> > | Learned Mean-Bound | __[99.9 ± 0.1]__ | __[99.9 ± 0.1]__ | *99.5 ± 0.2 | *99.4 ± 0.2 | 98.5 ± 0.7 | 98.4 ± 0.7 |
> > | Learned Max-Bound | 99.6 ± 0.5 | 99.6 ± 0.5 | 99.1 ± 0.3 | 99.1 ± 0.3 | 98.8 ± 0.7 | 98.8 ± 0.7 |
> >
> >
> > ---
> > > #### 2. Results on __HND__ (3L–64U)
> >
> > | Scheme | GAT Acc | GAT F1-ma | GCN Acc | GCN F1-ma | GraphSAGE Acc | GraphSAGE F1-ma |
> > |--------|---------:|-----------:|----------:|------------:|----------------:|------------------:|
> > | Complete Graph | *94.6 ± 1.2 | *94.5 ± 1.2 | *92.3 ± 3.5 | *92.3 ± 3.5 | 91.0 ± 4.0 | 91.0 ± 4.0 |
> > | Sentence Order | 92.6 ± 2.3 | 92.6 ± 2.3 | 88.7 ± 3.5 | 88.7 ± 3.5 | 91.1 ± 4.0 | 91.1 ± 4.0 |
> > | Window Co-occurrence | 92.1 ± 2.9 | 92.1 ± 2.9 | 88.3 ± 2.4 | 88.2 ± 2.4 | 93.5 ± 0.5 | 93.5 ± 0.5 |
> > | Mean Semantic Similarity | 91.2 ± 4.9 | 91.1 ± 5.0 | 90.8 ± 3.8 | 90.8 ± 3.8 | *93.6 ± 1.9 | *93.6 ± 1.9 |
> > | Max Semantic Similarity | 92.8 ± 5.5 | 92.8 ± 5.6 | 91.8 ± 3.5 | 91.8 ± 3.5 | 92.4 ± 3.9 | 92.4 ± 4.0 |
> > | Learned Mean-Bound | __[95.0 ± 2.2]__ | __[94.9 ± 2.2]__ | 86.0 ± 3.4 | 85.9 ± 3.4 | 93.1 ± 3.4 | 93.1 ± 3.4 |
> > | Learned Max-Bound | 92.6 ± 5.6 | 92.6 ± 5.6 | __92.5 ± 1.7__ | __92.5 ± 1.7__ | __94.9 ± 1.1__ | __94.8 ± 1.1__ |
> >
> > ---
> > > #### 3. Results on __arXiv__ (3L–64U)
> >
> > | Scheme | GAT Acc | GAT F1-ma | GCN Acc | GCN F1-ma | GraphSAGE Acc | GraphSAGE F1-ma |
> > |--------|---------:|-----------:|----------:|------------:|----------------:|------------------:|
> > | Sentence Order | 87.8 ± 0.5 | 87.3 ± 0.5 | 85.5 ± 0.8 | *84.9 ± 0.9 | 85.9 ± 0.7 | 85.3 ± 0.7 |
> > | Window Co-occurrence | 87.9 ± 1.9 | 87.4 ± 2.0 | __85.8 ± 0.8__ | __85.1 ± 0.9__ | 87.0 ± 1.0 | 86.4 ± 1.1 |
> > | Max Semantic Similarity | 87.8 ± 0.8 | 87.3 ± 0.8 | *85.6 ± 1.0 | *84.9 ± 1.1 | 86.3 ± 0.9 | 85.7 ± 1.0 |
> > | Learned Mean-Bound (sample) | *88.2 ± 3.1 | *87.6 ± 3.2 | 84.1 ± 1.5 | 83.5 ± 1.5 | *88.4 ± 0.9 | *87.9 ± 1.1 |
> > | Learned Max-Bound (sample) | __[91.7 ± 1.9]__ | __[91.3 ± 2.1]__ | 83.5 ± 1.6 | 82.9 ± 1.6 | __89.9 ± 1.4__ | __89.6 ± 1.3__ |

---

### Author Response · Authors · 2025-12-03
**Summary of Revisions Made in the Updated Manuscript**

We sincerely thank all reviewers for their thoughtful feedback and for the time invested in evaluating our work. In response, we have revised and expanded the manuscript. Below we summarize the key improvements introduced in the updated version, also referenced in our point-by-point rebuttals:

* **Clearer articulation of novelty and contributions (Section 1):** We revised the introduction to more explicitly position our work within the landscape of graph-based document classification, detailing how our graph-learning method and statistical filtering (sparsification) differ from both heuristic constructions and prior GSL approaches.

* **Clarification of limitations and distinction between heuristic-based and GSL methods (Section 2.2):** We expanded this section to more clearly delineate the shortcomings of heuristic document graphs and to accurately contextualize how our method fits within (and differs from) the broader GSL literature.

* **Refined methodological description (Section 3, Section 3.2, Appendix B):** We now explicitly describe how final attention weights are aggregated across heads, including the motivation for head averaging. We also added an illustrative figure (Figure 3) for improved clarity.

* **Addition of new non-graph baselines (Sections 4.2 and 4.3, Table 2):** To strengthen the empirical evaluation and ensure comparability across datasets, we incorporated: i) two transformer baselines: `Longformer-base` (4096-token context) and `LongT5-tglobal-base` (16k-token context), and ii) two simple but informative embedding-based baselines: `SentenceEmb + MLP` and `ReLUAttn-SentenceEmb + MLP`, allowing direct assessment of whether document structure provides benefits beyond strong semantic encoders alone.

* **Extended evaluation across multiple GNN backbones (Appendix C, Table 5):** We now report results using `GCN, GAT`, and `GraphSAGE`, demonstrating that the performance gains of our learned graphs persist across various architectures--highlighting the generality of our graph induction method.

* **Expanded discussion of results (Section 5):** We integrated the new baselines into the results analysis, offering a more comprehensive interpretation of our method.

* **New analyses on model convergence and learned attention distributions (Appendix B.2 and Appendix B.3):** We included training curves for the self-attention model, clarified early-stopping behavior, and added raw and averaged across heads attention visualizations to substantiate the behavior of our graph-learning proposal.


These revisions substantially strengthen the clarity, transparency, and empirical rigor of the paper. The improvements address the reviewers’ concerns and help more accurately reflect the contribution and impact of our work.

---

### Note · Authors · 2026-01-06

**Comment:**

We have decided to withdraw this submission from ICLR in order to pursue publication at another venue that aligns more closely with the scope of the work. We are grateful to the reviewers and area chairs for their time and thoughtful feedback, which has deeply helped us to improve the paper.

**Withdrawal Confirmation:**

I have read and agree with the venue's withdrawal policy on behalf of myself and my co-authors.